# NuTrea: Neural Tree Search
# for Context-guided Multi-hop KGQA

**Hyeong Kyu Choi** [*]
Computer Sciences
University of Wisconsin-Madison
hyeongkyu.choi@wisc.edu

**Seunghun Lee**
Computer Science & Engineering
Korea University
llsshh319@korea.ac.kr

**Jaewon Chu**
Computer Science & Engineering
Korea University
allonsy07@korea.ac.kr

**Hyunwoo J. Kim** [†]
Computer Science & Engineering
Korea University
hyunwoojkim@korea.ac.kr

## Abstract

Multi-hop Knowledge Graph Question Answering (KGQA) is a task that involves retrieving nodes from a knowledge graph (KG) to answer natural language questions. Recent GNN-based approaches formulate this task as a KG path searching problem, where messages are sequentially propagated from the seed node towards the answer nodes. However, these messages are past-oriented, and they do not consider the full KG context. To make matters worse, KG nodes often represent proper noun entities and are sometimes encrypted, being uninformative in selecting between paths. To address these problems, we propose Neural Tree Search (NuTrea), a tree search-based GNN model that incorporates the broader KG context. Our model adopts a message-passing scheme that *probes* the unreached subtree regions to boost the past-oriented embeddings. In addition, we introduce the Relation Frequency–Inverse Entity Frequency (RF-IEF) node embedding that considers the global KG context to better characterize ambiguous KG nodes. The general effectiveness of our approach is demonstrated through experiments on three major multi-hop KGQA benchmark datasets, and our extensive analyses further validate its expressiveness and robustness. Overall, NuTrea provides a powerful means to query the KG with complex natural language questions. Code is available at https://github.com/mlvlab/NuTrea.

## 1 Introduction

The knowledge graph (KG) is a multi-relational data structure that defines entities in terms of their relationships. Given its enormous size and complexity, it has long been a challenge to properly query the KG via human languages [1–6]. A corresponding machine learning task is knowledge graph question answering (KGQA), which entails complex reasoning on the KG to retrieve the nodes that correctly answers the given natural language question. To resolve the task, several approaches focused on parsing the natural language to a KG-executable form [7–10], whereas others tried to process the KG so that answer nodes can be ranked and retrieved [11–14, 2]. Building on these works, there has been a recent stream of research focusing on answering more complex questions with intricate constraints, which demand multi-hop reasoning on the KG.

---

[*]Work done at Korea University.

[†]Corresponding author.

37th Conference on Neural Information Processing Systems (NeurIPS 2023).

Answering complex questions on the KG requires processing both the KG nodes (entities) and edges (relations). Recent studies have addressed multi-hop KGQA by aligning the question text with KG edges (relations), to identify the correct path from seed nodes (*i.e.*, nodes that represent the question subjects) towards answer nodes. Many of these methods, however, gradually expand the search area outwards via message passing, whose trailing path information is aggregated onto the nodes, resulting in node embeddings that are past-oriented. Also, as many complex multi-hop KGQA questions require selecting nodes that satisfy specific conditions, subgraph-level (subtree-level) comparisons are necessary in distinguishing the correct path to the answer node. Furthermore, KG node entities often consist of uninformative proper nouns, and sometimes, for privacy concerns, they may be encrypted [15]. To address these issues, we propose **Neu**ral **Tr**ee **Sea**rch (NuTrea), a graph neural network (GNN) model that adopts a tree search scheme to consider the broader KG contexts in searching for the path towards the answer nodes.

NuTrea leverages expressive message passing layers that propagate subtree-level messages to explicitly consider the complex question constraints in identifying the answer node. Each message passing layer consists of three steps, Expansion → Backup → Node Ranking, whose Backup step *probes* the unreached subtree regions to boost the past-oriented embeddings with future information. Moreover, we introduce the Relation Frequency–Inverse Entity Frequency (RF-IEF) node embedding, which takes advantage of the global KG statistics to better characterize the KG node entities. Overall, NuTrea provides a novel approach in addressing the challenges of querying the KG, by allowing it to have a broader view of the KG context in aligning it with human language questions. The general effectiveness of NuTrea is evaluated on three major multi-hop KGQA benchmark datasets: WebQuestionsSP [16], ComplexWebQuestions [17], and MetaQA [18].

Then, our contributions are threefold:

- We propose *Neural Tree Search* (NuTrea), an effective GNN model for multi-hop KGQA, which adopts a tree search scheme with an expressive message passing algorithm that refers to the future-oriented subtree contexts in searching paths towards the answer nodes.

- We introduce *Relation Frequency-Inverse Entity Frequency* (RF-IEF), a simple node embedding technique that effectively characterizes uninformative nodes using the global KG context.

- We achieve the state-of-the-art on multi-hop KGQA datasets, WebQuestionsSP and ComplexWebQuestions, among weakly supervised models that do not use ground-truth logical queries.

## 2   Related Works

A knowledge graph (KG) is a type of a heterogeneous graph [19, 20] $\mathcal{G} = (\mathcal{V}, \mathcal{E})$, whose edges in $\mathcal{E}$ are each assigned to a relation type by the mapping function $f : \mathcal{E} \to \mathcal{R}$. The KGs contain structured information from commonsense knowledge [15, 21] to domain-specific knowledge [22], and the Knowledge Graph Question Answering (KGQA) task aims to answer natural language questions grounding on these KGs by selecting the set of answer nodes. Recent methods challenge on more complex questions that require multi-hop graph traversals to arrive at the correct answer node. Thus, the task is aliased as multi-hop KGQA or complex KGQA, which is generally discussed in terms of two mainstream approaches [23]: Semantic Parsing and Information Retrieval.

**Semantic Parsing.**   The main idea of semantic parsing-based methods is to first parse natural language questions into a logical query. The logical query is then grounded on the given KG in an executable form. For example, [8] applies the semantic query graph generated by natural language questions. In replace of hand-crafted query templates, [7] introduced a framework that automatically learns the templates from the question-answer pairs. Also, [24] proposed a novel graph generation method for query structure prediction. Other methods take a case-based reasoning (CBR) approach, where previously seen questions are referenced to answer a complex question. Approaches like [9, 25, 26] use case-based reasoning by referring to similar questions or KG structures. Recently, [10] proposed a framework that jointly infers logical forms and direct answers to reduce semantic errors in the logical query. A vast majority of methods that take the semantic parsing approach utilize the ground-truth logical forms or query executions during training. Thus, these supervised methods are generally susceptible to incomplete KG settings, but have high explainability.

**Information Retrieval.**   Information Retrieval-based methods focus on processing the KG to retrieve the answer nodes. The answer nodes are selected by ranking the subgraph nodes conditioned on the given natural language question. One of the former works [27] proposes an enhanced Key-Value

Memory neural network to answer more complex natural language questions. [28] and [29] extract answers from question-specific subgraphs generated with text corpora. To deal with incompleteness and sparsity of KG, [11] presents a KG embedding method to answer questions. Also, [12] handles this problem by executing query in the latent embedding space. In an effort to improve explainability of the information retrieval approach, [13] infers an adjacency matrix by learning the activation probability of each relation type. By adapting the teacher-student framework with the Neural State Machine (NSM), [14] made learning more stable and efficient. Furthermore, [30] utilized multi-task learning that jointly trains on the KG completion task and multi-hop KGQA, while [2] provides a novel link prediction framework. Recently, there has been a trend of borrowing the concept of logical queries from Semantic Parsing approaches, attempting to *learn* the logical instructions that guide the path search on the KG [2, 1]. Our NuTrea also builds on these approaches.

## 3 Method

In recent studies, models that sequentially process the knowledge graph (KG) from the seed nodes have shown promising results on the KGQA task. Building upon this approach, we propose Neural Tree Search (NuTrea). NuTrea adopts a novel message passing scheme that propagates the broader subtree-level information between adjacent nodes, which provides more context in selecting nodes that satisfy the complex question constraints. Additionally, we introduce a node embedding technique called *Relation Frequency–Inverse Entity Frequency* (RF-IEF), which considers the global KG information when initializing node features. These methods allow for a richer representation of each node by leveraging the broader KG context in answering complex questions on the KG.

### 3.1 Problem Definition

Here, we first define the problem settings for Neural Tree Search (NuTrea). The multi-hop KGQA task is primarily a natural language processing problem that receives a human language question $x_q$ as input, and requires retrieving the set of nodes from $\mathcal{G} = (\mathcal{V}, \mathcal{E})$ that answer the question. Following the standard protocol in KGQA [11], the subject entities in $x_q$ are given and assumed always to be mapped to a node in $\mathcal{V}$ via entity-linking algorithms [31]. These nodes are called seed nodes, denoted $v_s \in \mathcal{V}_s$, which are used to extract a subgraph $\mathcal{G}_q = (\mathcal{V}_q, \mathcal{E}_q)$ from $\mathcal{G}$ so that $\mathcal{V}_q$ is likely to contain answer nodes $v_a \in \mathcal{V}_a$. Then, the task reduces to a binary node classification problem of whether each node $v \in \mathcal{V}_q$ satisfies $v \in \mathcal{V}_a$.

Following prior works [14, 1], we first compute different question representation vectors with a language model based module, called Instruction Generators (IG). We have two IG modules, each for the Expansion (section 3.2.1) and Backup (section 3.2.2) step, to compute

$$\{\mathbf{q}_{\text{exp}}^{(i)}\}_{i=1}^N = \text{IG}_{\text{exp}}(x_q), \quad \{\mathbf{q}_{\text{bak}}^{(j)}\}_{j=1}^M = \text{IG}_{\text{bak}}(x_q), \tag{1}$$

where we name $\mathbf{q}_{\text{exp}}^{(i)}, \mathbf{q}_{\text{bak}}^{(j)} \in \mathbb{R}^D$ as the expansion instruction and backup instruction, respectively. Detailed description on the IG module is in the supplement. Then, the learnable edge (relation) type embeddings $\boldsymbol{R} \in \mathbb{R}^{|\mathcal{R}| \times D}$ are randomly initialized or computed with a pretrained language model. On the other hand, node embeddings $\boldsymbol{H} \in \mathbb{R}^{|\mathcal{V}_q| \times D}$ of $\mathcal{V}_q$ are initialized using the edges in $\mathcal{E}_q$ and their relation types by function $\mathcal{H}$ as $\boldsymbol{H} = \mathcal{H}(\mathcal{E}_q, \boldsymbol{R})$ (*e.g.*, $\mathcal{H} \equiv$ arithmetic mean of incident edge relations). This is because the node entities are often uninformative proper nouns or encrypted codes. Then, our NuTrea model $\mathcal{F}$ function is defined as

$$\hat{\mathbf{y}} = \mathcal{F}(\{\mathbf{q}_{\text{exp}}^{(i)}\}_{i=1}^N, \{\mathbf{q}_{\text{bak}}^{(j)}\}_{j=1}^M, \mathcal{V}_s ; \mathcal{G}_q, \boldsymbol{H}, \boldsymbol{R}), \tag{2}$$

where $\hat{\mathbf{y}} \in \mathbb{R}^{|\mathcal{V}_q|}$ is the predicted node score vector normalized across nodes in $\mathcal{V}_q$, whose ground-truth labels are $\mathbf{y} = [\mathbb{1}(v \in \mathcal{V}_a)]_{v \in \mathcal{V}_q} \in \mathbb{B}^{|\mathcal{V}_q|}$. Then, the model is optimized with the KL divergence loss between $\hat{\mathbf{y}}$ and $\mathbf{y}$. In this work, we claim our contributions in $\mathcal{F}$ (section 3.2) and $\mathcal{H}$ (section 3.3).

### 3.2 Neural Tree Search ($\mathcal{F}$)

Our Neural Tree Search (NuTrea) model consists of multiple layers that each performs message passing in three consecutive steps: (1) Expansion, (2) Backup, and (3) Node ranking. The Expansion step propagates information outwards to expand the search tree, which is followed by the Backup step, where the depth-$K$ subtree content is aggregated to each of their root nodes, to enhance the past-oriented messages from the Expansion step. Then, the nodes are scored based on how likely they answer the given question.

Figure 1: **Neural Tree Search.** Given a natural language question, a corresponding KG subgraph $\mathcal{G}_q$ is extracted, and expansion instructions $\mathbf{q}_{exp}$ and backup instructions $\mathbf{q}_{bak}$ are computed with Instruction Generators (IG). The node embeddings of $\mathcal{G}_q$ are first defined by our RF-IEF (section 3.3), which characterizes nodes based on the global KG information by suppressing prevalent relation types that hold less meaning. Then, in each NuTrea layer (section 3.2), messages are propagated outwards from the seed node with respect to $\mathbf{q}_{exp}$ (Expansion), which contain future-oriented subtree information conditioned by $\mathbf{q}_{bak}$ (Backup). Then, the nodes are scored by adding and normalizing the logits (Node Ranking), which are utilized in the subsequent layer. The figure describes the first layer of NuTrea whose Backup subtree depth $K$ is 2, and $\phi$ is the softmax function. Overall, NuTrea considers the broader KG contexts in distinguishing the correct paths, contrary to previous methods that do not incorporate the Backup step.

### 3.2.1 Expansion

Starting from the seed node $v_s \in \mathcal{V}_s$, a NuTrea layer first expands the search tree by sequentially propagating messages outwards to the adjacent nodes. The propagated messages $\mathbf{f}_{uv}^{(i)}$ are conditioned on the expansion instructions $\{\mathbf{q}_{exp}^{(i)}\}_{i=1}^{N}$, which are computed as

$$\mathbf{f}_{uv}^{(i)} = \text{ReLU}(W_{\mathbf{f}}\mathbf{r}_{uv} \odot \mathbf{q}_{exp}^{(i)}), \tag{3}$$

where $i \in [1, N]$, $W_{\mathbf{f}} \in \mathbb{R}^{D \times D}$ is a learnable linear projection, $\mathbf{r}_{uv} = \text{row}(\mathbf{R}) \in \mathbb{R}^D$ is the relation type embedding of edge $u \to v$, and $\odot$ is an element-wise product operator. Optionally, we use the relative position embedding $\mathbf{e}_{uv} \in \mathbb{R}^D$ as

$$\mathbf{f}_{uv}^{(i)} = \text{ReLU}((W_{\mathbf{f}}\mathbf{r}_{uv} + \mathbf{e}_{uv}) \odot \mathbf{q}_{exp}^{(i)}), \tag{4}$$

where $\mathbf{e}_{uv}$ is defined for each relation type. Then, for an edge $u \to v$, $N$ types of messages are propagated, which are element-wise products between the edge relation and the $N$ different question representation vectors. This operation highlights the edges that are relevant to the given question. Then, the messages are aggregated to a node $v$ via an MLP aggregator, computed as

$$\tilde{\mathbf{f}}_v = \Big\|_{i=1}^{N} \sum_{u \in \mathcal{N}(v)} s_u \mathbf{f}_{uv}^{(i)}$$

$$\mathbf{f}_v = \text{MLP}(\mathbf{h}_v \| \tilde{\mathbf{f}}_v), \tag{5}$$

where $\mathbf{h}_v = \text{row}(\mathbf{H}) \in \mathbb{R}^D$ is the node embedding and $s_u \in \mathbb{R}$ is the score value of a head node $u$. In the first layer, the seed nodes are the only head nodes, whose scores are initially set to 1. In subsequent layers, we use the updated node scores as $s_u$, whose computation will be introduced shortly in section 3.2.3. Notably, the nodes with score $s_u = 0$, which are typically nodes that are yet to be reached, do not pass any message to their neighbors.

### 3.2.2 Backup

After the Expansion step grows the search tree, the leaf nodes of the tree naturally contain the trailing path information from the seed nodes, which is past-oriented. To provide future context, we employ

the Backup step to aggregate contextual information from subtrees rooted at the nodes reached by previous NuTrea layers. We denote a subtree of depth $K$ rooted at node $v$ as $\mathcal{T}_v^K = (\mathcal{V}_v, \mathcal{E}_v) \subset \mathcal{G}_q = (\mathcal{V}_q, \mathcal{E}_q)$. Here, $\mathcal{V}_v = \{u \mid SP(u, v) \leq K \text{ and } u \in \mathcal{V}_q\}$ and $\mathcal{E}_v = \{(u_1, u_2) \mid (u_1, u_2) \in \mathcal{E} \text{ and } u_1, u_2 \in \mathcal{V}_v\}$, where $SP(u, v)$ is a function that returns the length of the shortest path between nodes $u$ and $v$. For the Backup step, we consider only the edge set $\mathcal{E}_v$ of $\mathcal{T}_v^K$. The reason behind this is that the edges (relation types) better represent the question context in guiding the search on the KG. Also, using both the node and edge sets may introduce computational redundancy, as the initial node features originate from the edge embeddings [14, 1] (See section 3.1).

To pool the constraint information from $\mathcal{E}_v$, we apply max-pooling conditioned on the question content. Specifically, we take a similar measure as the Expansion step by computing $M$ types of messages conditioned on the backup instructions $\{\mathbf{q}_{\text{bak}}^{(j)}\}_{j=1}^M$.

$$\mathbf{c}_{u_1 u_2}^{(j)} = \text{ReLU}(W_{\mathbf{c}} \mathbf{r}_{u_1 u_2} \odot \mathbf{q}_{\text{bak}}^{(j)}), \tag{6}$$

where $j \in [1, M]$ and $W_{\mathbf{c}} \in \mathbb{R}^{D \times D}$. Then, we max-pool the messages as

$$\mathbf{c}_v^{(j)} = \text{MAX-POOL}(\{\mathbf{c}_{u_1 u_2}^{(j)} \mid (u_1, u_2) \in \mathcal{E}_v\}), \tag{7}$$

which represents the extent to which the local subtree context of $v$ is relevant to the conditions and constraints in the question. Next, the information is aggregated with an MLP layer, and the node embedding $\mathbf{h}_v$ is updated as

$$\begin{aligned} \mathbf{c}_v &= \Big\|_{j=1}^M \mathbf{c}_v^{(j)} \\ \mathbf{h}_v &:= \text{MLP}(\mathbf{f}_v \| \mathbf{c}_v), \end{aligned} \tag{8}$$

where $\mathbf{f}_v$ refers to the propagated embeddings originating from Eq. (5). This serves as a correction of the original past-oriented message with respect to question constraints, providing the next NuTrea layer with rich local context.

### 3.2.3 Node Ranking

Finally, each node is scored and ranked based on the embeddings before and after the Backup step. For node $v$, $\mathbf{f}_v$ from Eq. (5) and $\mathbf{h}_v$ from Eq. (8) are projected to the expansion-score $s_v^{(e)}$ and backup-score $s_v^{(b)}$, respectively, as

$$s_v^{(e)} = \mathbf{f}_v \cdot W_e \qquad s_v^{(b)} = \mathbf{h}_v \cdot W_b, \tag{9}$$

where $W_e, W_b \in \mathbb{R}^D$. The final node score is retrieved by adding the two scores. We use a context coefficient $\lambda$ to control the effect of the Backup step, and apply softmax to normalize the scores as

$$s_v = \text{Softmax}([s_v^{(e)} + \lambda \cdot s_v^{(b)}])_{v \in \mathcal{V}_q}, \tag{10}$$

which is passed on to the next layer to be used for Eq. (5). Figure 1 provides a holistic view of our method, and pseudocode is in the supplement.

Overall, the message passing scheme of NuTrea resembles the algorithm of Monte Carlo Tree Search (MCTS): Selection → Expansion → Simulation → Backup. The difference is that our method replaces the node 'Selection' and 'Simulation' steps with a soft GNN-based approach that rolls out subtrees and updates the nodes at once, rather than applying Monte Carlo sampling methods.

### 3.3 RF-IEF Node Embedding ($\mathcal{H}$)

Another notable challenge with KGs is embedding nodes. Many KG entities (nodes) are proper nouns that are not informative, and several KGQA datasets [16, 17] consist of encrypted entity names. Thus, given no proper node features, the burden is on the model layers to learn meaningful node embeddings. To alleviate this, we propose a novel node embedding method, Relation Frequency-Inverse Entity Frequency (RF-IEF), which grounds on the local and global topological information of the KG nodes.

Term frequency-inverse document frequency (TF-IDF) is one effective feature that characterizes a textual document [32–34]. The bag-of-words model computes the frequency of terms in a document and penalizes frequent but noninformative words, *e.g.,* 'a', 'the', 'is', 'are', by the frequency of the

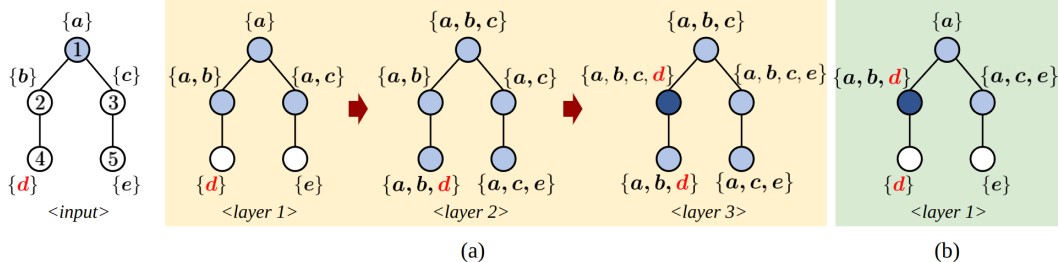

Figure 2: **Expressiveness of NuTrea layers.** In the input KG subgraph (left-most figure), let node 1 and 2 be the seed and answer node, respectively, where information $d$ is critical in choosing node 2 as the answer. The shaded nodes indicate the regions that are sequentially reached by consecutive GNN layers. Compared to (a) previous methods [14, 1] that require 3 message passing steps for node 2 to access $d$, our (b) NuTrea layer requires only 1 step for node 2 to acquire information $d$ to determine it as the answer.

term across documents. Motivated by the idea, we represent a node on a KG as a bag of relations. An entity node is characterized by the frequencies of rare (or informative) relations. Similar to TF-IDF, we define two functions: Relation Frequency (RF) and Inverse Entity Frequency (IEF). The RF function is defined for node $v \in \mathcal{V}_q$ and relation type $r \in \mathcal{R}$ as

$$\text{RF}(v, r) = \sum_{e \in \mathcal{I}(v)} \mathbb{1}\{f(e) = r\}, \tag{11}$$

where $\mathcal{I}(v)$ is the set of incident edges of node $v$, $\mathbb{1}$ is an indicator function, and $f : \mathcal{E} \to \mathcal{R}$ is a function that retrieves the relation type of an edge. Then, the output of the RF function is a matrix $RF \in \mathbb{R}^{|\mathcal{V}_q| \times |\mathcal{R}|}$ that counts the occurrence of each relation type incident to each node in the KG subgraph. We used raw counts for relation frequency (RF) to reflect the local degree information of a node. On the other hand, the IEF function is defined as

$$\text{IEF}(r) = \log \frac{|\mathcal{V}_q|}{1 + \text{EF}(r)}, \tag{12}$$

where

$$\text{EF}(r) = \sum_{v \in \mathcal{V}_q} \mathbb{1}\{\exists\, e \in \mathcal{I}(v) \text{ s.t. } f(e) = r\}. \tag{13}$$

EF counts the global frequency of nodes across KG subgraphs that have relation $r$ within its incident edge set. With $IEF \in \mathbb{R}^{|\mathcal{R}|}$, the RF-IEF matrix $\boldsymbol{F} \in \mathbb{R}^{|\mathcal{V}_q| \times |\mathcal{R}|}$ is computed as

$$\boldsymbol{F} = RF \, \text{diag}(IEF), \tag{14}$$

where $\text{diag}(IEF) \in \mathbb{R}^{|\mathcal{V}_q| \times |\mathcal{R}|}$ denotes a diagonal matrix constructed with the elements of $IEF$.

The RF-IEF matrix $\boldsymbol{F}$ captures both the local and global KG structure and it can be further enhanced with the rich semantic information on the edges of KGs. Unlike entity nodes with uninformative text, *e.g.,* proper nouns and encrypted entity names, edges generally are accompanied by linguistic descriptions (relation types). Hence, the relations are commonly embedded by a pre-trained language model in KGQA. Combining with the relation embeddings $\boldsymbol{R} \in \mathbb{R}^{|\mathcal{R}| \times D}$, our final RF-IEF node embeddings $\boldsymbol{H}$ is computed as

$$\boldsymbol{H} = \boldsymbol{F} \, \boldsymbol{R} \, \boldsymbol{W}_h, \tag{15}$$

where $\boldsymbol{H} \in \mathbb{R}^{|\mathcal{V}_q| \times D}$, and $\boldsymbol{W}_h \in \mathbb{R}^{D \times D}$. The RF-IEF node embeddings can be viewed as the aggregated semantics of relations, which are represented by a language model, based on graph topology as well as the informativeness (or rareness) of relations. A row in $\boldsymbol{H}$ is a node embedding vector $\mathbf{h}_v$, which is used in Eq. (5) at the first NuTrea layer.

## 3.4 Discussion

**Expressiveness of the NuTrea layer.** While many previous multi-hop KGQA models *simultaneously* update all node embeddings, recent works [14, 2, 1] have shown the superiority of approaches that search paths on the KG. These methods gradually expand the searching area by *sequentially*

updating nodes closer to the seed node towards the answer nodes. Our model builds on the latter *sequential search* scheme as well, enhancing expressiveness with our proposed NuTrea layers.

With a simple toy example in Figure 2, we compare the message flow of previous sequential search models and our NuTrea. The example demonstrates that our NuTrea layer (b) can probe subtrees to quickly gather fringe node (*i.e.*, node 4 or 5) information without exhaustively visiting them. This is accomplished by our Backup step, which boosts the original past-oriented node embeddings with future-oriented subtree information.

## 4 Experiments

### 4.1 Experimental Settings

**Datasets.** We experiment on three large-scale multi-hop KGQA datasets: MetaQA [35], WebQuestionsSP (WQP) [16] and ComplexWebQuestions (CWQ) [17]. Meta-QA consists of three different splits, 1-hop, 2-hop, and 3-hop, each indicating the number of hops required to reach the answer node from the seed node. The questions are relatively easy with less constraints. WQP and CWQ, on the other hand, contain more complex questions with diverse constraints. WQP is relatively easier, as CWQ is derived from WQP by extending its questions with additional constraints. MetaQA is answerable with the WikiMovies knowledge base [27], while WQP and CWQ require the Freebase KG [15] to answer questions. Further dataset information and statistics are provided in the supplement.

**Baselines.** We mainly compare with previous multi-hop KGQA methods that take the Information Retrieval approach (section 2). These models, unlike Semantic Parsing approaches, do not access the ground truth logical queries and focus on processing the KG subgraph to rank the nodes to identify answer nodes. To introduce the three most recent baseline models: (1) SQALER [3] proposes a scalable KGQA method whose complexity scales linearly with the number of relation types, rather than nodes or edges. (2) TERP [2] introduces the rotate-and-scale entity link prediction framework to integrate textual and KG structural information. (3) ReaRev [1] adaptively selects the next reasoning step with a variant of breadth-first search (BFS). Other baselines are introduced in the supplement.

**Implementation Details.** For WQP, 2 NuTrea layers with subtree depth $K = 1$ is used, while CWQ with more complex questions uses 3 layers with depth $K = 2$. In the case of RF-IEF node embedding, we pre-compute the Entity Frequency (EF) values in Eq. (13) for subgraphs in the training set before training. We use the same EF values throughout training, validation, and testing. This stabilizes computation by mitigating the large variance induced by relatively small batch sizes. For MetaQA, the number of NuTrea layers are selected from $\{2, 3\}$, and $K$ for ego-graph pooling from $\{1, 2\}$. See the supplement for further hyperparameter settings and details.

### 4.2 Main Experiments

Here, we present the experimental results of NuTrea. Following the common evaluation practice of previous works, we test the model that achieved the best performance on the validation set. In the WQP dataset experiments in Table 1, we achieved the best performance of 77.4 H@1 among strong KGQA baselines that take an information retrieval approach, as discussed in Section 2. Compared to the previous best, this is a large improvement of 0.6 points. In terms of the F1 score, which evaluates the answer *set* prediction, our method achieved a score of 72.7, exceeding the previously recorded value by a large margin of 1.8 points. In addition, we also improved the previous state-of-the-art performance on the CWQ dataset by achieving an 53.6 H@1, which is an improvement of 0.7 points.

We also experimented NuTrea on MetaQA to see if it performs reasonably well with easy questions as well. On three data splits, NuTrea achieved comparable performance with previous state-of-the-art methods for simple question answering. Evaluating with the average H@1 score of the three splits, NuTrea performs second best among all baseline models.

### 4.3 Incomplete KG Experiments

The KG is often human-made and the contents are prone to being incomplete. Hence, it is a norm to test a model's robustness on incomplete KG settings where a certain portion of KG triplets, *i.e.*, a tuple of ⟨head, relation, tail⟩, are dropped. This experiment evaluates the robustness of our model to missing relations in a KG. We follow the experiment settings in [1], and use the identical incomplete KG dataset which consists of WQP samples with [50%, 30%, 10%] of the original KG triplets remaining. In Table 2, NuTrea performs the best in most cases, among the GNN models

Table 1: **Results on multi-hop KGQA datasets.** The Hit@1 and F1 scores are reported. The baselines are taken from the original papers. The best performances are in bold, and the second best are underlined.

| Models | WQP | | CWQ | | MetaQA | | | |
|---|---|---|---|---|---|---|---|---|
| | H@1 | F1 | H@1 | F1 | 1-hop | 2-hop | 3-hop | Avg. H@1 |
| KV-Mem [27] | 46.7 | 38.6 | 21.1 | - | 95.8 | 25.1 | 10.1 | 43.7 |
| GraftNet [28] | 66.7 | 62.4 | 32.8 | - | - | - | - | 96.8 |
| PullNet [29] | 68.1 | - | 45.9 | - | 97.0 | 99.9 | 91.4 | 96.1 |
| EmbedKGQA [11] | 66.6 | - | - | - | **97.5** | 98.8 | 94.8 | 97.0 |
| ReifiedKB [36] | 52.7 | - | - | - | 96.2 | 81.1 | 72.3 | 83.2 |
| EMQL [37] | 75.5 | - | - | - | 97.2 | 98.6 | 99.1 | 98.3 |
| TransferNet [13] | 71.4 | 48.6 | 48.6 | - | **97.5** | **100.0** | **100.0** | **99.2** |
| NSM(+p) [14] | 73.9 | 66.2 | 48.3 | 44.0 | 97.3 | 99.9 | 98.9 | 98.7 |
| NSM(+h) [14] | 74.3 | 67.4 | 48.8 | 44.0 | 97.2 | 99.9 | 98.9 | 98.6 |
| Rigel [38] | 73.3 | - | 48.7 | - | - | - | - | - |
| SQALER+GNN [3] | 76.1 | - | - | - | - | 99.9 | 99.9 | - |
| TERP [2] | 76.8 | - | 49.2 | - | **97.5** | 99.4 | 98.9 | 98.6 |
| ReaRev [1] | 76.4 | 70.9 | 52.9 | - | - | - | - | - |
| **NuTrea (Ours)** | **77.4** | **72.7** | **53.6** | **49.5** | 97.4 | **100.0** | 98.9 | 98.8 |

Table 2: **Incomplete KG experiments.** NuTrea also performs well in incomplete KG settings. The baseline figures were taken from [1].

| Portion of KG triplets (%) | 50% | | 30% | | 10% | |
|---|---|---|---|---|---|---|
| | H@1 | F1 | H@1 | F1 | H@1 | F1 |
| Graftnet [28] | 47.7 | 34.3 | 34.9 | 20.4 | 15.5 | 6.5 |
| SGReader [39] | 49.2 | 33.5 | 35.9 | 20.2 | 17.1 | 7.0 |
| HGCN [40] | 49.3 | 34.3 | 35.2 | 21.0 | 18.3 | 7.9 |
| ReaRev [1] | 53.4 | 39.9 | 37.9 | 23.6 | **19.4** | 8.6 |
| **NuTrea (Ours)** | **53.7** | **40.1** | **38.3** | **24.1** | 18.9 | **8.7** |

Table 3: **Component ablation experiments.** The ablation experiments were done on the WQP dataset. Two main contributions are studied.

| RF-IEF Node Emb. | Backup step | WQP H@1 | WQP F1 |
|---|---|---|---|
| ✓ | ✓ | **77.4** (–0.0) | **72.7** (–0.0) |
| ✓ | | 74.8 (–2.6) | 70.4 (–2.3) |
| | ✓ | 76.8 (–0.6) | 71.5 (–1.2) |
| | | 73.4 (–4.0) | 70.9 (–1.8) |

designed to handle incomplete KGs. We believe that our model adaptively learns multiple alternative reasoning processes and can plan for future moves beforehand via our Backup step, so that it provides robust performance with noisy KGs.

## 5 Analysis

In this section, we provide comprehensive analyses on the contributions of NuTrea to ensure its effectiveness in KGQA. We try to answer the following research questions: **Q1.** How does each component contribute to the performance of NuTrea? **Q2.** What is the advantage of NuTrea's tree search algorithm over previous methods? **Q3.** What are the effects of the RF-IEF node embeddings?

### 5.1 Ablation Study

Here, we evaluate the effectiveness of each component in NuTrea to answer **Q1**. An ablation study is performed on our two major contributions: the Backup step in our NuTrea layers and the RF-IEF node embeddings. By removing the RF-IEF node embeddings, we instead apply the common node initialization method used in [14, 1], which simply averages the relation embeddings incident to each node. Another option is to use zero embeddings, but we found it worse than the simple averaging method. For NuTrea layer ablation, we remove the Backup step which plays a key role in aggregating the future-oriented subtree information onto the KG nodes. Then, only the expansion-score ($s_v^{(e)}$) is computed, and the backup-score ($s_v^{(b)}$) is always 0. Also, the embedding $\mathbf{f}_v$ (Eq. (5)), is directly output from the NuTrea layer and no further updates are made via the Backup step.

In Table 3, we can see that the largest performance drop is observed when the Backup step is removed. Without it, the model has limited access to the broader context of the KG and cannot reflect the complex question constraints in node searching. Further discussion on this property of NuTrea's message passing is provided in the next Section 5.2. Also, we observed a non-trivial 0.6 point drop

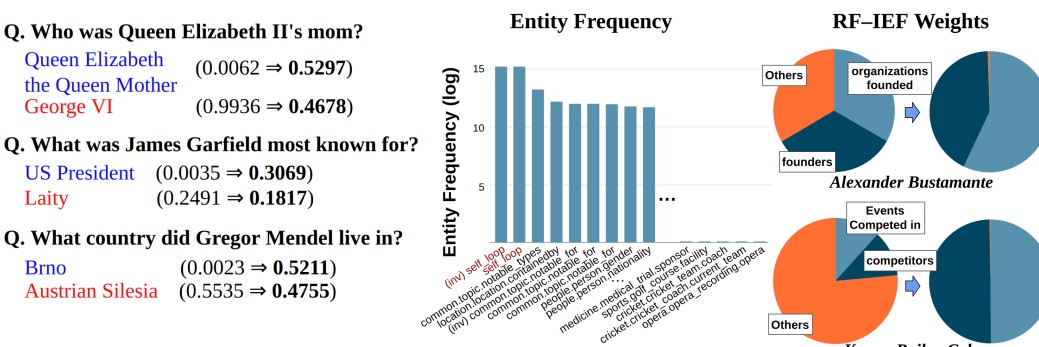

**Q. Who was Queen Elizabeth II's mom?**
Queen Elizabeth the Queen Mother (0.0062 ⇒ **0.5297**)
George VI (0.9936 ⇒ **0.4678**)

**Q. What was James Garfield most known for?**
US President (0.0035 ⇒ **0.3069**)
Laity (0.2491 ⇒ **0.1817**)

**Q. What country did Gregor Mendel live in?**
Brno (0.0023 ⇒ **0.5211**)
Austrian Silesia (0.5535 ⇒ **0.4755**)

Figure 3: **Qualitative examples (without Backup ⇒ with Backup)** (section 5.2.2).

Figure 4: **Weights computed by RF-IEF** (section 5.3).

in H@1 by removing the RF-IEF initialization method. By ablating both components, there was a significant degradation of 4.0 H@1 points.

## 5.2 Advantages of NuTrea

To answer **Q2**, we highlight the key advantages of NuTrea over recent approaches. We analyze the efficiency of NuTrea, and provide qualitative results.

### 5.2.1 Efficiency of NuTrea

In addition, to further verify the utility of our model, we provide analyses on the latency of NuTrea with WQP in Table 4. Compared to the most recent ReaRev [1] model, training and inference latency per epoch/sample is slightly bigger, due to our additional Backup module. However,

Table 4: **Latency of ReaRev and NuTrea.**

| Models | NuTrea (Ours) | ReaRev |
|---|---|---|
| Training Latency (per epoch) | 100.2 s | 78.0 s |
| Inference Latency (per sample) | 67.7 ms | 51.3 ms |
| Training GPU Hours | 2.9 H | 4.3 H |

thanks to our expressive message passing scheme, NuTrea converges way faster, allowing the training GPU hours to be reduced from 4.3 hours to 2.9 hours.

To provide more insight in terms of number of NuTrea layers, we also reveal its effect on model performance, in Figure 5. The figure reports the F1 scores for both with and without the Backup module, evaluated on the WebQuestionsSP dataset. "NuTrea without Backup" has an equivalent model configuration used in the Backup ablation experiment of Table 3. Overall, the performance of "NuTrea without Backup" does generally improve with additional layers, but "NuTrea with Backup" reached the highest score of 72.7 with only 2 NuTrea layers. This is enabled by our Backup module, which alleviates the burden of exhaustively searching deeper

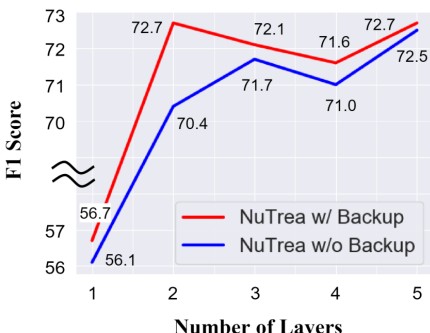

Figure 5: **Effect of number of layers.**

nodes. This is computationally more efficient than stacking multiple layers to achieve higher performance. Specifically, by comparing the 2-layer "NuTrea with Backup" and 5-layer "NuTrea without Backup", the former required an average of 73.8 ms of inference time per question, whereas the latter required 108.6 ms. With only 68% of compute, our NuTrea achieved comparable performance with the deeper "NuTrea without Backup". Note, these latency values were evaluated on a different environment from values reported in Table 4.

### 5.2.2 Qualitative Results

In Figure 3, we demonstrate several qualitative examples from our error analysis. In each question, the blue (node) entity is the correct answer choice, while the red is the wrong choice made by the

model without the Backup step. The values in parentheses demonstrate the difference in node scores between models *without* and *with* Backup. Without it, the sequential search model cannot refer to local contexts of a node and frequently predicts an extremely low score for the correct answer node (*e.g.*, 0.0062 for "Queen Elizabeth the Queen Mother" in the first question of Figure 3). Such a problem is mitigated by our NuTrea's Backup step, which tends to boost scores of correct answers and tone down wrong choices that were wrongly assigned with a high score. More qualitative examples are provided in the supplement, along with an analysis on the different importances of the Backup step in different datasets, by controlling the context coefficient $\lambda$.

### 5.3 Effect of RF-IEF

The RF-IEF node embedding is a simple method inspired by an effective text representation technique in natural language processing. Here, we disclose the specific effect of RF-IEF on the relation embedding aggregation weights, thereby answering **Q3**. In Figure 4 (left), the log-scaled $EF$ (Eq. 13) value of each relation type is sorted. The globally most frequent relation types, including "self_loop"s, are too general to provide much context in characterizing a KG node. Our RF-IEF suppresses such uninformative relation types for node embedding initialization, resulting in a weight distribution like Figure 4 (right). The pie charts display two examples on the difference between aggregation weights for a node entity before and after RF-IEF is applied. To illustrate, the weight after RF-IEF corresponds to a row of $F$ in Eq. (14), while the weight before RF-IEF would be uniform across incident edges. As "Alexander Bustamante" is a politician, the relation types "orgainizations founded" and "founders" become more lucid via RF-IEF, while relations like "events_competed_in" and "competitors" are emphasized for an athlete like "Kemar Bailey-Cole". Likewise, RF-IEF tends to scale up characteristic relation types in initializing the node features, thereby enhancing differentiability between entities. To further demonstrate RF-IEF's general applicability, we also provide a plug-in experiment on another baseline model in the supplement.

## 6 Conclusion

Neural Tree Search (NuTrea) is an effective GNN model for multi-hop KGQA, which aims to better capture the complex question constraints by referring to the broader KG context. The high expressiveness of NuTrea is attained via our message passing scheme that resembles the MCTS algorithm, which leverages the future-oriented subtree information conditioning on the question constraints. Moreover, we introduce the RF-IEF node embedding technique to also consider the global KG context. Combining these methods, our NuTrea achieves the state-of-the-art in two major multi-hop KGQA benchmarks, WebQuestionsSP and ComplexWebQuestions. Further analyses on KG incompleteness and the qualitative results support the effectiveness of NuTrea. Overall, NuTrea reveals the importance of considering the broader KG context in harnessing the knowledge graph via human languages.

## Acknowledgments and Disclosure of Funding

This work was partly supported by ICT Creative Consilience program (IITP-2023-2020-0-01819) supervised by the IITP, the National Research Foundation of Korea (NRF) grant funded by the Korea government (MSIT) (NRF-2023R1A2C2005373), and KakaoBrain corporation.

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
