# NuTrea: Neural Tree Search for Context-guided Multi-hop KGQA (Supplement)

**Overview.** We here provide additional materials that were excluded from the main paper due to limited space. We provide the pseudocode of our method in section A; The Instruction Generator is described in section B; Dataset details and statistics are summarized in section C; All baseline models are briefly introduced in section D; Implementation details and specific hyperparameter settings for each dataset are enlisted in section E; A statistical significance test is conducted on NuTrea in section F; The effect of the context coefficient $\lambda$ is analyzed to verify the effect of the Backup step in section G; Additional qualitative examples are provided in section H; NuTrea was compared with deeper baseline models to demonstrate its effectiveness in section I; RF-IEF is applied to another model to check its generality in section J; A short discussion on RF-IEF and Adamic-Adar is provided in section K; Limitations and possible future directions are finally discussed in section L.

## A    Pseudocode

---
**Algorithm 1** NuTrea Layer

---
**Input:** KG subgraph $\mathcal{G}_q = (\mathcal{V}_q, \mathcal{E}_q)$, Relation embeddings $\boldsymbol{R}$, Node embeddings $\mathbf{h}_{v \in \mathcal{V}_q} = \text{row}(\boldsymbol{H})$, Node scores $s_{u \in \mathcal{V}_q}$, Expansion instructions $\{\mathbf{q}_{\text{exp}}^{(i)}\}_{i=1}^{N}$, Backup instructions $\{\mathbf{q}_{\text{bak}}^{(j)}\}_{j=1}^{M}$

1: **for** $i = 1$ **to** $N$ **do**
2:     $\mathbf{f}_v^{(i)} \leftarrow \phi(\{\mathbf{f}_{uv} | u \in \mathcal{N}(v), \mathbf{q}_{\text{exp}}^{(i)}, \boldsymbol{R}, s_{u \in \mathcal{V}_q}\})$                     ▷ Expansion
3: **end for**
4: $\tilde{\mathbf{f}}_v \leftarrow \|_{i=1}^{N} \mathbf{f}_v^{(i)}$
5: $\mathbf{f}_v \leftarrow \text{MLP}(\mathbf{h}_v \| \tilde{\mathbf{f}}_v)$
6: **for** $j = 1$ **to** $M$ **do**
7:     $\mathbf{c}_v^{(j)} \leftarrow \text{MAX-POOL}(\{\mathbf{c}_{u_1 u_2} | (u_1, u_2) \in \mathcal{E}_v, \mathbf{q}_{\text{bak}}^{(j)}, \boldsymbol{R}\})$             ▷ Backup
8: **end for**
9: $\mathbf{c}_v \leftarrow \|_{j=1}^{M} \mathbf{c}_v^{(j)}$
10: $\mathbf{h}_v \leftarrow \text{MLP}(\mathbf{f}_v \| \mathbf{c}_v)$
11: $s_v^{(e)} \leftarrow \mathbf{f}_v \cdot W_e$
12: $s_v^{(b)} \leftarrow \mathbf{h}_v \cdot W_b$
13: $s_v \leftarrow \text{Softmax}([s_v^{(e)} + \lambda \cdot s_v^{(b)}])_{v \in \mathcal{V}_q}$                            ▷ Node Ranking
14: **return** $\mathbf{h}_v, s_v$

---

For better understanding, we here provide the pseudocode of our NuTrea layer in Algorithm 1. Each layer consists of (1) Expansion, (2) Backup, and (3) Node Ranking.

---
**Algorithm 2** Full Inference
---
**Input:** Question text $x_q$, KG subgraph $\mathcal{G}_q = (\mathcal{V}_q, \mathcal{E}_q)$, Initial score vector $\mathbf{s}_0 \in \mathbb{B}^{|\mathcal{V}_q|}$, Relation embeddings $\boldsymbol{R}$

1: $\boldsymbol{H}_0 \leftarrow \text{RF-IEF}(\mathcal{V}_q, \mathcal{E}_q, \boldsymbol{R})$
2: $\{\mathbf{q}_{\text{exp}}^{(i)}\}_{i=1}^{N} \leftarrow \text{IG}_{\text{exp}}(x_q)$
3: $\{\mathbf{q}_{\text{bak}}^{(j)}\}_{j=1}^{M} \leftarrow \text{IG}_{\text{bak}}(x_q)$
4: **for** $l = 1$ **to** $L$ **do**
5: $\quad \boldsymbol{H}_l, \mathbf{s}_l \leftarrow \text{NuTrea-Layer}_{(l)}(\boldsymbol{H}_{l-1}, \mathbf{s}_{l-1}, \{\mathbf{q}_{\text{exp}}^{(i)}\}_{i=1}^{N}, \{\mathbf{q}_{\text{bak}}^{(j)}\}_{j=1}^{M})$
6: **end for**
7: $\mathbf{q}_{\text{exp}} \leftarrow \text{expansion-instruction-update}(\mathbf{q}_{\text{exp}}, \boldsymbol{H}_L)$
8: $\mathbf{q}_{\text{bak}} \leftarrow \text{backup-instruction-update}(\mathbf{q}_{\text{bak}}, \boldsymbol{H}_L)$
9: $\boldsymbol{H}_0 \leftarrow \boldsymbol{H}_L$
10: **for** $l = 1$ **to** $L$ **do**
11: $\quad \boldsymbol{H}_l, \mathbf{s}_l \leftarrow \text{NuTrea-Layer}_{(l)}(\boldsymbol{H}_{l-1}, \mathbf{s}_{l-1}, \{\mathbf{q}_{\text{exp}}^{(i)}\}_{i=1}^{N}, \{\mathbf{q}_{\text{bak}}^{(j)}\}_{j=1}^{M})$
12: **end for**
13: **return** $\mathbf{s}_L$
---

We also provide pseudocode for the entire inference process in Algorithm 2, which includes some details that were not discussed in the main paper. Similar to [1], the inference process repeats the GNN forward pass multiple times to compute the final node score. However, we find iterating only twice is enough in our case, thanks to our effective RF-IEF node embedding. Also note that after the first forward pass, the expansion and backup instructions are modified before the next iteration.

## B   Instruction Generator Descriptions

In Neural Tree Search, we extract $N + M$ different question representations. $N$ are expansion instructions, while $M$ are backup instructions. The instruction vectors are computed with a module commonly used in KGQA (*e.g.*, NSM [2], ReaRev [1]) to retrieve $N$ different question representations. Specifically, we followed [1]. An IG takes the natural language question $\{\mathbf{q}_{\text{exp}}^{(i)}\}_{i=1}^{N}$ as input to output question representations as

$$\{\mathbf{q}_{\text{exp}}^{(i)}\}_{i=1}^{N} = \text{IG}_{\text{exp}}(x_q). \tag{1}$$

In the IG function, a tokenizer converts input $x_q$ to tokens $\{\mathbf{x}_t\}_{t=1}^{T}$ that are used to retrieve the sentence embedding $\mathbf{q}_{\text{LM}}$ with a language model $\text{LM}(\cdot)$ (e.g., SentenceBERT) as

$$\mathbf{q}_{\text{LM}} = \text{LM}(\{\mathbf{x}_t\}_{t=1}^{T}). \tag{2}$$

To deterministically sample a sequence of sentence representations, a quasi-monte carlo sampling (or non-probability sampling) approach is adopted. $\mathbf{q}_{\text{LM}}$ is first used to compute attention weights $a_t^{(i)}$ as

$$\mathbf{q}^{(i)} = \boldsymbol{W}^{(i)}(\mathbf{q}^{(i-1)}||\mathbf{q}_{\text{LM}}||\mathbf{q}_{\text{LM}} - \mathbf{q}^{(i-1)}||\mathbf{q}_{\text{LM}} \odot \mathbf{q}^{(i-1)}) \tag{3}$$

$$a_t^{(i)} = \text{Softmax}(\boldsymbol{W}_a(\mathbf{q}^{(i)} \odot \mathbf{x}_t)), \tag{4}$$

where $i \in [1, N]$, and $\mathbf{q}^{(0)}$ is a zero vector. Also, $||$ indicates the concatenation operator, and $\boldsymbol{W}_a \in \mathbb{R}^{D \times D}$, $\boldsymbol{W}_a \in \mathbb{R}^{D \times D}$ are learnable matrices. Finally, each question representation $\mathbf{q}_{\text{exp}}^{(i)}$ is computed as

$$\mathbf{q}_{\text{exp}}^{(i)} = \sum_t a_t^{(i)} \mathbf{x}_t. \tag{5}$$

The same process is repeated to compute the Backup instructions $\{\mathbf{q}_{\text{bak}}^{(i)}\}_{i=1}^{M}$.

## C   Dataset Details

Here, we provide further details regarding the benchmark datasets.

**WebQuestionsSP**    contains 4,727 natural language questions that can be answered with the Freebase KG [3]. For each question, at least one topic entity is assigned and a subgraph within 2-hops from the topic entities is extracted. Approximately 30% of the questions require more than two KG triplets to attain the correct answers, and 7% of them need complex reasoning with the constraints in the question [1]. Also, note that there are two evaluation protocols. One uses a validation set with size 100, which was adopted in [4]. Another has a validation set size of 250, which originates from [5]. We follow the latter setting for our experiments.

**ComplexWebQuestions**    contains 34,689 natural language questions also answerable with Freebase KG. This dataset is derived from WQP by extending the questions with additional constraints. The questions require more complex reasoning with composition, conjunction, comparison, and superlatives. Accordingly, up to 4-hops are needed to reach an answer node.

**MetaQA**    contains more than 400K questions, which consists of three splits: 1-hop, 2-hop, and 3-hop. Each split's questions are answered by traversing the corresponding amount of hops on the KG extracted from the WikiMovies [6] knowledge base. The KG incorporates information on the movie domain, and contains 43K entities (nodes), 9 relation types (predicates), and 135K triplets (facts).

Table 1 contains additional dataset statistics for each train, validation, test split.

Table 1: **Dataset statistics.**

| Datasets | Train | Val | Test |
|---|---|---|---|
| WebQuestionsSP | 2,848 | 250 | 1,639 |
| ComplexWebQuestions | 27,639 | 3,519 | 3,531 |
| MetaQA 1-hop | 96,106 | 9,992 | 9,947 |
| MetaQA 2-hop | 118,948 | 14,872 | 14,872 |
| MetaQA 3-hop | 114,196 | 14,274 | 14,274 |

## D    Other Baselines

We here provide full introduction of the 10 baselines we compared with NuTrea in the main table:

(1) KV-Mem [6] enhances the key-value memory network, (2) GraftNet [5] is a GNN-based model, (3) PullNet [7] is also a GNN-based model that attempts to generate a more question-relevant subgraph, (4) EmbedKGQA [4] attempts to embed the knowledge graph entities and select the answer node based on the similarity between the entity embeddings and the question embedding. (5) EMQL [8] and Rigel [9] attempts to enhance ReifiedKB [10]. (6) TransferNet [11] aims to enhance explainability of KGQA reasoning by explicitly learning the transition matrix for each reasoning step. (7) NSM [2] is a GNN-based model that adopts the Neural State Machine, (8) SQALER [12] proposes a scalable KGQA method whose complexity scales linearly with the number of relation types, rather than nodes or edges. (9) TERP [13] introduces the rotate-and-scale entity link prediction framework to integrate textual and KG structural information. (10) ReaRev [1] adaptively selects the next reasoning step with a variant of breadth-first search (BFS).

## E    Hyperparameters and Experimental Details

To train our models, we use a single RTX 3090 GPU with a RAdam optimizer [14]. The initial learning rate is set to 0.0005 with an exponential learning rate scheduler with a decay rate of 0.99. Also, we use the KL divergence as the loss function. To evaluate the F1 score, we retrieve nodes in the order that has higher confidence until the total probability sums up to 0.95. Other hyperparameter settings that are used for the experiments are tabularized in Table 2.

## F    Statistical Significance Test

During training, we have observed that the models' performance variance is quite high. Thus, it would be more reasonable to compare the performances by evaluating the average of multiple runs to validate the statistical significance of the gains. As WebQuestionsSP showed the highest variance, we

Table 2: **Hyperparameter settings.**

| Hyperparameter | WQP | CWQ | MetaQA 1-hop | MetaQA 2-hop | MetaQA 3-hop |
|---|---|---|---|---|---|
| batch size | 8 | 8 | 32 | 32 | 32 |
| train epochs | 100 | 50 | 10 | 10 | 10 |
| model dimension | 100 | 100 | 50 | 50 | 50 |
| max-pool depth $K$ | 1 | 2 | 1 | 1 | 2 |
| context coefficient $\lambda$ | 0.3 | 1.0 | 1.0 | 1.0 | 1.0 |
| relative positional embedding | O | X | X | O | O |
| Expansion instruction num. $N$ | 2 | 3 | 2 | 2 | 2 |
| Backup instruction num. $M$ | 3 | 3 | 3 | 3 | 3 |
| layer num. $L$ | 2 | 3 | 2 | 3 | 3 |
| dropout probability | 0.3 | 0.3 | 0.2 | 0.2 | 0.2 |

conducted a t-test on the multiple runs with five different seeds (0,1,2,3,4) to assess the statistical significance of our NuTrea over comparable models. Unfortunately, however, only the source code for ReaRev is currently available among the three comparable models in our table (e.g. SQALER, TERP, ReaRev). So we compare NuTrea only with ReaRev.

In Table 3, we report the "average (std dev)" of WQP experiments along with the t-test p-value. Our NuTrea's average H@1 across five seeds excels ReaRev by 2.1, and by 2.0 for average F1 score. To evaluate the statistical significance of the differences, we applied the t-test and retrieved p-values of 0.020 and 0.009, respectively. Thus, we can assure that the performance gain is statistically significant. The trained parameters will be released along with the code.

Table 3: **t-test results on ReaRev and NuTrea.**

| Models | ReaRev | NuTrea (Ours) | t-test p-value |
|---|---|---|---|
| Average H@1 | 74.2 (1.4) | 76.3 (1.4) | 0.020* |
| Average F1 | 69.8 (1.2) | 71.8 (0.7) | 0.009** |

## G  Sensitivity Analysis on the Context Coefficient

Here, we analyze the context coefficient $\lambda$ that balances the effect of the Backup step. Figure 1 shows the H@1 and F1 scores of our models trained on WQP and CWQ with different context coefficients $\lambda \in [0.3, 0.6, 1.0]$ As mentioned in the main paper, the CWQ dataset contains more complex questions that require intricate consideration of diverse constraints compared to the WQP dataset. As we expected, on CWQ the models with a relatively larger context coefficient $\lambda$ perform better, which means that the constraint information needs to be further considered with the Backup step. For WQP dataset with simpler questions, the model trained with a small $\lambda$ achieves the overall best performance. These experimental results evidence that our model can be optimized depending on the datasets by properly choosing context coefficient $\lambda$ to balance the impact of the Backup module.

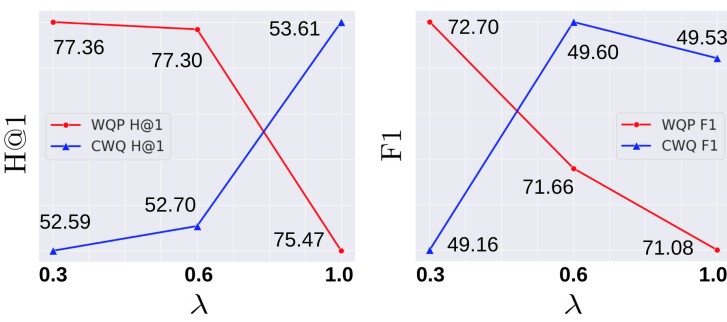

Figure 1: **Effect of the context coefficient** $\lambda$. The NuTrea model trained on the CWQ with more complex questions prefers larger $\lambda$ values, whereas the WQP models prefer relatively smaller coefficients.

## H    Additional Qualitative Examples

Quantitatively, the Backup step had a significant impact on the H@1 and F1 score by improving them from 74.8 to 77.4, and 70.4 to 72.7, respectively (See Table 3 of main paper). Here, in Figure 2, we extend the list of qualitative examples provided in the main paper, to support the effectiveness of our Backup module. Similar to the main paper, the blue corresponds to the correct answer node entity, while red is the wrong choice made by the model that does not leverage the Backup step in message passing. The values in the parentheses refer to the scores of ("Without Backup" ⇒ "With Backup").

**Q. Who was queen Isabella's mother?**

Isabella of Portugal, Queen of Castile    (0.4399 ⇒ **0.5439**)          John II of Castile    (0.5601 ⇒ **0.4560**)

**Q. Where is Olympic national park located?**

Washington    (0.5000 ⇒ **0.5114**)          Jefferson County    (0.5000 ⇒ **0.4874**)

**Q. What was Hitler the leader of?**

Nazi Party    (0.0709 ⇒ **0.7023**)          German Workers' Party    (0.6411 ⇒ **0.2534**)

**Q. What is the most populated city in the United States?**

New York    (0.0116 ⇒ **0.6484**)          Vermont    (0.3250 ⇒ **0.0353**)

**Q. Who was vice president for Lincoln?**

Andrew Johnson    (0.0026 ⇒ **0.4197**)          Hannibal Hamlin    (0.5127 ⇒ **0.4067**)

**Q. What was Jesse James killed with?**

Firearm    (0.4873 ⇒ **0.5259**)          Assassination    (0.5127 ⇒ **0.4709**)

**Q. Who is the leader of North Korea today?**

Kim Jong-un    (0.3091 ⇒ **0.4454**)          Kim Il-sung    (0.3602 ⇒ **0.4755**)

Figure 2: **More Qualitative Examples.** The blue choices are the correct answers, whereas red choices are the wrongly selected answers by the model without Backup.

## I    Model Scale Analysis

Table 4: **Comparison with ReaRev of different scales.**

| Model | # Layers | # Params | H@1 | F1 |
|---|---|---|---|---|
| ReaRev-2 (base) | 2 | 23.47 M | 75.4 | 70.4 |
| ReaRev-3 | 3 | 23.49 M | 74.0 | 69.9 |
| ReaRev-4 | 4 | 23.50 M | 73.8 | 69.5 |
| ReaRev-5 | 5 | 23.51 M | 74.4 | 70.5 |
| ReaRev-5-wide | 5 | 27.86 M | 75.0 | 70.3 |
| NuTrea (ours) | 2 | 27.43 M | **77.3** | **72.2** |

In order to verify whether our NuTrea has an advantage over a deeper ReaRev [1] model variants, we provide relevant experimental results in Table 4. We tested a deeper model with 2 to 5 layers, and also tried increasing the model dimension, denoted ReaRev-5-wide. Notably, ReaRev-5-wide has more parameters compared to our NuTrea model. In the table, NuTrea outperforms all ReaRev variants by significant margins. Also note that all models were trained under a new identical environment.

We conjecture the key advantage of NuTrea over deeper ReaRev is that it has a smaller smoothing effect. Assume, for instance, that the model is at a node that is 2 hops away from the seed node (*i.e.*,

on the 2nd layer) and needs to be confirmed by the depth-4 information. NuTrea requires only 2 expansion steps (message-passing layers), and 2 backup steps without any node/edge updates. The baseline model, on the other hand, will need up to 6 layers (i.e., hops) to achieve the same effect.

## J    Generality of RF-IEF

To verify how RF-IEF node embedding improves baseline models, we also applied our method to ReaRev [1]. The benchmark setting adopts the node initialization method used in [2], as

$$e^{(0)} = \sigma \left( \sum_{<e',r,'e> \in \mathcal{N}_e} r \cdot W_T \right), \tag{6}$$

where $e^{(0)}$ is the initialized embedding of a node, $r$ is the relation embedding, and $W_T$ is a learnable matrix. This equation does not reweight relation types and applies a uniform weight over relation embeddings. In contrast, our RF-IEF highlights distinctive relation types to better characterize the node entity. In Table 5, the average H@1 performance over multiple runs (seeds $0 \sim 4$) of ReaRev and NuTrea are improved by 0.4 and 0.8, respectively, when our RF-IEF embeddings are applied.

Table 5: **RF-IEF applied to ReaRev and NuTrea.**

| Models | ReaRev | NuTrea (Ours) |
|---|---|---|
| without RF-IEF | 74.2 (1.4) | 75.5 (1.1) |
| with RF-IEF | **74.6** (0.8) | **76.3** (1.4) |

## K    Analogy of RF-IEF to Adamic-Adar

Adamic-Adar [15] is a popular measure to quantify the linkage between two nodes in a graph. The Adamic-Adar function $A(x, y)$ is defined as

$$A(x, y) = \sum_{u \in \mathcal{N}(x) \cap \mathcal{N}(y)} \frac{1}{\log |\mathcal{N}(u)|}, \tag{7}$$

which measures the sum of the inverse logarithmic degree value of the shared neighbors for two arbitrary nodes $x$ and $y$. That is, the function counts the number of neighboring nodes, scaled disproportionally to each of the neighbors' popularity (*i.e.*, how central the neighbor is, measured by its degree value). This aspect of Adamic-Adar shares an intuition with our Relation Frequency-Inverse Entity Frequency (RF-IEF) node embedding technique. Similar to Admaic-Adar, which scales down neighbor *nodes* that are connected everywhere, RF-IEF suppresses omnipresent *relation* types in defining the node via the IEF function. Considering that Adamic-Adar-style methods have long been successful in many tasks that deal with defining and predicting linkages [16–18], the idea of initializing a node in a similar fashion is promising and intuitive.

## L    Limitations and Future Work

We observed that quite a great portion of the questions in WebQuestionsSP and ComplexWebQuestions do not contain an answer node in the extracted subgraph. Our model does not specifically deal with such cases, and sometimes outputs predictions with high probability even when no answer node exists. To our knowledge, no research has tackled this problem regarding confidence calibration or out-of-distribution KG settings. Calibrating the confidence score and enabling the model to reasonably predict a null set will be an important future research direction.