# OpenReview forum: "NuTrea: Neural Tree Search for Context-guided Multi-hop KGQA"
_NeurIPS.cc/2023/Conference — NeurIPS 2023 poster_

### Official Review · Reviewer_5UYV · 2023-07-03

**Soundness:** 3 good
**Presentation:** 3 good
**Contribution:** 2 fair
**Rating:** 5
**Confidence:** 4

**Summary:**

This paper proposes a tree search-based GNN model that contains three consecutive steps: expansion, backup, and node ranking. It also proposes a relation frequency-inverse entity frequency node embedding method. The proposed model achieves new SOTA on WebQSP and CWQ datasets.

**Strengths:**

1. It proposes a tree-search-based GNN method with backup propagation to extend the KG context, which is intuitive and novel.
2. The RF-IEF node embedding, motivated by TF-IDF, is also a novel method and experimentally effective.
3. The proposed method achieves SOTA on WebQSP and CWQ datasets.
4. Extensive experiments including incomplete KG setting, ablation study, and advantage analysis of the proposed method.

**Weaknesses:**

1. The enhancement relative to the leading benchmark approach, ReaRev, is marginal, notably on the CWQ dataset. The progress is even more slight in the context of the incomplete KG scenario, as highlighted in Table 2.

2. In such circumstances, a more detailed comparative analysis with ReaRev might be required. ReaRev, which employs a variation of the breadth-first search for adaptive reasoning, also strives to expand the modeling of the KG context. How is the proposed tree search method better than ReaRev? More crucially, given that RF-IEF supplements the GNN model, what could be the outcome if RF-IEF were integrated with ReaRev? Would it still perform worse than NuTrea?

**Questions:**

In the ablation study in Table 3, we see a significant performance drop when not using the Backup step. Under this setting, have the authors tried varying the layers of GNN? It would be interesting to see the performance-#layer curves with and without the backup step.

**Limitations:**

Yes

---

> ### Author Rebuttal · Authors · 2023-08-06
>
> ### Q1. Performance gains are marginal on ComplexWebQuestions and incomplete KG experiments.
> NuTrea’s 0.7 H@1 gain aligns with the standard improvements observed in recently published works. For instance, TERP (COLING 2022) reported a 0.6 H@1 gain compared to the previous best, Rigel (EMNLP 2021) reported a 0.1 H@1 gain, and TransferNet (EMNLP 2021) lags behind NSM (WSDM 2021) by 0.2 H@1. Therefore, when evaluated within the context of these existing works, our 0.7 point gain is far from marginal.
>
> For incomplete KG experiments, on the other hand, it is important to note that incomplete KG experiments involve training with a considerably low portion of KG triplets (10%, 30%, and 50%). As a result, the performance level in such a setting reaches a state of high saturation, making any improvement even more noteworthy.
>
> &nbsp;
>
> ### Q2. How is the proposed tree search method better than ReaRev?
> The key advantage of NuTrea from ReaRev is the Backup module that considers the subtree contents (i.e., the unreached regions), without explicitly visiting or updating child nodes in the subtree. This approach is crucial in distinguishing between correct and incorrect node choices, especially when the same sequence of relations (i.e., meta paths) can lead to both correct and incorrect nodes. This enhancement allows NuTrea to make more informed decisions during the tree search process based on broader subtree-level contexts, leading to overall performance improvement.
>
> &nbsp;
>
> ### Q3. If RF-IEF were integrated with ReaRev, would it still perform worse than NuTrea?
> In fact, a plug-in experiment on ReaRev is already provided in Table 4 of Appendix I. We brought the table below for your convenience. In the table, we observed that ReaRev’s average performance improves with RF-IEF, but does not yet match that of NuTrea. Note, the values in parentheses are standard deviations across 5 runs.
> | Models         |   ReaRev   | NuTrea (Ours) |
> |----------------|:----------:|:-------------:|
> | without RF-IEF | 74.2 (1.4) |   75.5 (1.1)  |
> | with RF-IEF    | 74.6 (0.8) |   76.3 (1.4)  |
>
> &nbsp;
>
> ### Q4. The performance-#layer curves of NuTrea with and without the Backup step.
> We have added the performance-#layer curve figure in the attached PDF file above in the "Author Rebuttal by Authors" thread. The figures report the WebQuestionsSP F1 score of our NuTrea with and without the Backup module. We observed that the model with Backup consistently outperforms its counterpart. Also, the model without Backup does generally improve with more GNN layers, but NuTrea without Backup reached the highest score with only 2 layers.

---

> > ### Comment · Reviewer_5UYV · 2023-08-17
> > **Question about the Backup step**
> >
> > Thanks for the rebuttal. But I'm still confused about the importance of the Backup step. In your answer to Q4, NuTrea w/o backup can achieve competitive performance when scaling to 5 layers, which means that simply adding the number of layers can bring performance gain, hindering the importance of the backup step. If the authors want to claim that adding layers will increase the latency, please provide the latencies of NuTrea w/ backup (2 layers) and NuTrea w/o backup (5 layers).

---

> > > ### Author Response · Authors · 2023-08-17
> > > **Additional Comment by the Authors**
> > >
> > > Absolutely! The following table compares the latency per training epoch of two NuTrea configurations. But please note that the latencies were measured in a new environment, and these values are not to be compared with the latency reported in Table 4 of our manuscript.
> > >
> > > | Model Config.             | Train Latency (sec/epoch) |
> > > |---------------------------|:-------------------------:|
> > > | 2-layer NuTrea w/ Backup (base) |           653.2           |
> > > | 5-layer NuTrea w/o Backup |           1068.0          |
> > >
> > > The "5-layer Nutrea without Backup" requires approximately 64% more training time than our original model, "2-layer NuTrea with Backup". Thus, we'd like to claim that Backup is a much more efficient method for aggregating depth information, compared to naively increasing the number of layers.

---

> > > > ### Comment · Reviewer_5UYV · 2023-08-18
> > > > **More about the backup step**
> > > >
> > > > Thank you for sharing the training latency. Can you also provide the inference latency? Moreover, including a theoretical analysis would strengthen this paper.

---

> > > > > ### Author Response · Authors · 2023-08-18
> > > > > **Additional Comment by the Authors**
> > > > >
> > > > > Below is the full table that includes inference latency as well. Inference time is 47% slower for the 5-layer NuTrea without Backup.
> > > > >
> > > > > | Model Config.             | Train Latency (sec/epoch) | Inference Latency (ms/sample) |
> > > > > |---------------------------|:-------------------------:|:-------------------------:|
> > > > > | 2-layer NuTrea w/ Backup (base) |           653.2           |  73.8 |
> > > > > | 5-layer NuTrea w/o Backup |           1068.0          | 108.6 |
> > > > >
> > > > > Also, thank you very much for your suggestion. We will try to provide a theoretical analysis in our final version, as it will require a careful investigation and verification process. Thanks again!

---

### Official Review · Reviewer_QeEo · 2023-07-06

**Soundness:** 4 excellent
**Presentation:** 3 good
**Contribution:** 3 good
**Rating:** 6
**Confidence:** 4

**Summary:**

In this paper, a retrieval-based solution to the Multi-hop KGQA problem is proposed. The authors introduce a novel approach that incorporates two innovations: a GNN network with bidirectional message passing (Expansion and Backup) and a novel method for embedding knowledge graph vertices, inspired by TF-IDF.

The paper features numerous experiments and comparisons with existing baselines using well-known datasets such as WebQuestionsSP, ComplexWebQuestions, and Meta-QA, all of which rely on Freebase KG.
Furthermore, the authors provide an extensive analysis of the proposed technique, showing that the introduced backup method makes the maximum contribution to the improvement of the results.


**Strengths:**

In this paper, the authors introduce a new method to the solution of the KGQA problem. The backup component of their solution stands out as particularly intriguing because of its maximum contribution to the improvement of the final results. The method is explained in a clear manner, with all essential formulas described qualitatively. Explanation of the inner working of the method is clear. NuTrea, the proposed approach, demonstrates results that are comparable to the State-of-the-Art (SOTA). Additionally, the authors have conducted qualitative comparisons with existing methods. Openly releasing the code will  benefit the community.


**Weaknesses:**

The authors did not compare their proposed method with papers that showed significantly better results, such as DECAF (DPR + FiD-3B) (Yu, et al., 2023) with a Hit@1 of 78.8. The reason for not doing so was not explained.
One aspect of the system utilizes graph vertex embeddings, and the authors introduce a new approach called RF-IEF. However, it is unclear why the authors did not compare the performance of NuTrea with other graph node embeddings like PyTorch-BigGraph, which may have yielded better results. Currently, it is difficult to evaluate the effectiveness of the proposed vertex embedding approach.


**Questions:**

Experiments conducted on Incomplete KG demonstrate that the proposed approach performs notably better compared to others. However, it is not specified in the paper whether RF-IEF was re-trained separately for each Incomplete KG or if the original one was used. If the latter case is true, the experiment is unfair and should be recalculated.

**Limitations:**

NuTrea limits the search for answers to the vicinity of the original vertex, which restricts the depth of exploration within the graph. The initial depth of search is predetermined and additional heuristics are required.

---

> ### Author Rebuttal · Authors · 2023-08-06
>
> ### Q1. Why is NuTrea not compared with other methods (e.g. DECAF) that have better results?
> Among the various KGQA methods, the ones that outperform NuTrea belong to the “Semantic Parsing” category, where they leverage ground-truth logical queries (forms) during training (refer to section 2, Related Works). The logical queries are like ground-truth paths to the answer nodes, which can substantially enhance model performance through supervised training. On the contrary, NuTrea and its baselines belong to the "Information Retrieval" category, where direct supervision using ground truth logical queries is not utilized. To our knowledge, NuTrea demonstrates the best performance among the Information Retrieval approaches.
>
> &nbsp;
>
> ### Q2. Why is RF-IEF not compared with other graph node embeddings like PyTorch-BigGraph?
> We compared RF-IEF with three commonly used node embeddings for knowledge graphs, namely TransE, DistMult, and ComplEx. These node embeddings were trained solely on the training set, similar to how we computed the RF-IEF only on the train data. As shown in the table below, the results demonstrate that NuTrea with RF-IEF significantly outperforms the other node embeddings. We conjecture that the main reason for this is that other node embeddings were not contrived for the inductive settings in the KGQA task. That is, node entities that were not seen during the embedding training steps had to be initialized to zero vectors, which would have inevitably damaged performance. Our RF-IEF (and also the previously used method from NSM [1]), on the other hand, is better suited for inductive settings.
>
> In the case of PyTorch-BigGraph, the pretrained embeddings were labeled with different ID’s, preventing us from mapping the embeddings to the corresponding nodes.
>
> | KG Node Embeddings | WQP H@1 | WQP F1 |
> |:------------------:|:-------:|:------:|
> |       TransE       |   74.6  |  70.0  |
> |      DistMult      |   75.5  |  69.9  |
> |       ComplEx      |   74.1  |  68.7  |
> |       NSM [1]      |   76.8  |  71.5  |
> |       **RF-IEF**  (ours)     |   **77.4**  |  **72.7**  |
>
> [1] Gaole He, Yunshi Lan, Jing Jiang, Wayne Xin Zhao, and Ji-Rong Wen. Improving multi-hop knowledge base question answering by learning intermediate supervision signals. In WSDM, 2021.
>
> &nbsp;
>
> ### Q3. Was RF-IEF retrained for each incomplete KG experiment?
> Yes, the RF-IEF was recomputed for each incomplete KG experiment, and no pretrained parameters were used for RF-IEF. Thus, the KG incompleteness experiment in Table 2 of the manuscript is valid.

---

### Official Review · Reviewer_jk9R · 2023-07-06

**Soundness:** 3 good
**Presentation:** 3 good
**Contribution:** 3 good
**Rating:** 7
**Confidence:** 4

**Summary:**

The paper presented two improvements to graph-based (KG) QA tasks: (1) Backup step, and (2) RF-IEF. Both proposed methods, according to Table 3, lead to significant improvement on multiple multi-hop QA tasks.

**Strengths:**

The proposed idea of including a BackUp step, which resembles MTCS, is very interesting. Experimentally, it leads to the biggest difference (> 2 points) on the WQ dataset. To my knowledge, this is first paper which explicitly write out the backup term, while previous research relies on the GCN to capture contextual information from the neighbors.

The RF-IEF is also an interesting observation. The improvement, however, is not as significant. As mentioned by the authors, "Many KG entities (nodes) are pronouns that are not informative, and several KGQA datasets [11, 12] consist of encrypted entity names." Can you please run your model on a subset of questions with "pronouns" and/or encrypted entity names" to justify this claim?

**Weaknesses:**

Overall, this is a good paper.

Please consider running experiments suggested above, i.e. with "pronouns" and/or encrypted entity names".

Other comments:
1. Line 105, it's a bit unfair to say "Following the standard protocol in KGQA [5], the subject entities in 𝑥𝑞 are given and assumed always to be mapped to a node in V". There are many work which doesn't make that assumption. It would be better to say this is an assumption you made and make sure that numbers in Table 1, 2, 3 all follow this assumption.

2. Can you please specify what are "edge type" in line 116?

**Questions:**

1. Can you please specify your loss function? Is the model trained end-to-end? Any intermediate supervision?
2. Can you please explain why the ablation study is performed on WQ? Some questions in WQ are 1-hop questions. CWQ should have more multi-hop questions where BackUp should benefit more.

**Limitations:**

Please see weakness/questions above.

---

> ### Author Rebuttal · Authors · 2023-08-06
>
> ### Q1. Can you run your model on a subset of questions with "pronouns" and/or encrypted entity names"?
> All the questions in WebQuestionsSP and ComplexWebQuestions entail encrypted KG entities. Thus, the results in our paper already represent the experimental setting in question.
>
> &nbsp;
>
> ### Q2. Is "Following the standard protocol in KGQA, the subject entities in $x_q$ are given and assumed always to be mapped to a node in $\mathcal{V}$" the authors’ assumption?
> This is not an assumption we made on our own. It is is a standard protocol of KGQA datasets, whose seed nodes (subject entities) are always provided along with the KG subgraph for each question. Thus, we can assume that this assumption holds as long as we evaluate our model on WebQuestionsSP, ComplexWebQuestions, or MetaQA.
>
> &nbsp;
>
> ### Q3. what is "edge type" in line 116?
> In this work, “edge type” is equivalent to “relation type”. The edge type (relation type) defines the factual relationship between the two connected nodes (entities) on the KG.
>
> &nbsp;
>
> ### Q4. Can you specify your loss function? Is the model trained end-to-end? Any intermediate supervision?
> Yes, the model is trained end-to-end without any intermediate supervision. As specified in Line 122~123, we use the KL divergence loss between the predicted score $\mathbf{\hat{y}} \in \mathbb{R}^{|\mathcal{V}\_q|}$ and the ground truth multi-hot vector $\mathbf{y}  \in \mathbb{B}^{|\mathcal{V}\_q|}$.  That is,
>
> $\mathbf{\hat{y}} = \text{NuTrea}(\\{\mathbf{q}\_{\text{exp}}^{(i)}\\}\_{i=1}^N, \\{\mathbf{q}\_\text{bak}^{(j)}\\}\_{j=1}^M, \mathcal{V}\_s)$\
> $\mathcal{L} = \text{KLD}(\mathbf{\hat{y}}, \mathbf{y}),$
>
> where $\\{\mathbf{q}\_{\text{exp}}^{(i)}\\}\_{i=1}^N$ refers to the expansion instructions, $ \\{\mathbf{q}\_\text{bak}^{(j)}\\}\_{j=1}^M$ refers to the backup instructions, $\mathcal{V}\_s$ is the set of seed nodes, and $\text{KLD}(\cdot)$  is the KL divergence function.
>
> &nbsp;
>
> ### Q5. Ablation study on CWQ.
> The ablation results on ComplexWebQuestions are provided in the table below.
> | RF-IEF | Backup | CWQ H@1 | CWQ F1 |
> |:------:|:------:|:-------:|:------:|
> |    v   |    v   |   53.6  |  49.5  |
> |    v   |        |   52.7  |  48.7  |
> |        |    v   |   53.1  |  47.4  |
> |        |        |   51.7  |  48.7  |

---

### Official Review · Reviewer_E45E · 2023-07-07

**Soundness:** 2 fair
**Presentation:** 1 poor
**Contribution:** 2 fair
**Rating:** 6
**Confidence:** 3

**Summary:**

The proposed model adopts a message-passing scheme that probes the unreached subtree regions to boost node embeddings.The work also introduces the Relation Frequency–Inverse Entity Frequency (RF-IEF) node embedding that considers the global KG context to better characterize ambiguous KG nodes. The method shows some effectiveness over multiple datasets.

**Strengths:**

-	Good results on multiple datasets
-	RF-IEF style node embeddings introduced. Might be useful in other works too.
-	Nice Analysis section.

**Weaknesses:**

-	Model sizes/ compute not clearly discussed. Some information present in Table 4 but insufficient.
-	How is the backup step different from a bigger expansion set in the previous works. This model doesn’t bring in any significant inductive bias over previous models.
 1. The backup is like a DFS during the expansion BFS. Really depends on the dataset as to where you want to focus your compute on. One could either explore or exploit.
 2. How would this model holdup in 4 or 5 hop QA? Would it not be extremely inefficient?
-	The gains are somewhat marginal and might just be achieved by increasing parameters/expansion depth in previous models.
-	RF-IEF is pitched as a generalized node embedding method but was not tested in other models. Although some nice analysis shown.
-	Bad writing. Lot of forward references to future text - Eq 1, Eq 5.
-	IG model descriptions very handwavy in paper and appendix.

**Questions:**

-	Are relation and relation type the same thing?
-	Can you list situations where this backup not useful? Or is it always useful?
-	Also address the weaknesses

**Limitations:**

The authors discuss some limitations.

---

> ### Author Rebuttal · Authors · 2023-08-06
>
> ### Q1. Model size comparison is needed.
> Below is the table containing the number of model parameters of NuTrea and ReaRev. Although NuTrea contains more parameters, it requires far less training hours than ReaRev. Also, increasing the model size of ReaRev does not necessarily enhance performance (See Q2).
>
> |  | Parameters | Training GPU Hours |   |   |
> |:------:|:----------:|:------------------:|---|---|
> | NuTrea |50 M| 2.9 Hrs | | |
> | ReaRev|24 M| 4.3 Hrs | | |
>
> &nbsp;
>
> ### Q2. Can performance gain be achieved by increasing parameters or expansion depth in previous models?
> No significant performance improvement was observed in the previous model, ReaRev, by simply increasing the expansion depth (i.e., the number of layers). Various expansion depths were tested, ranging from 2 to 5, but the H@1 and F1 performances did not improve much with deeper models. In fact, the original model with 2 layers performed better than deeper variants. Additionally, an attempt was made to enhance the ReaRev-5 model's width by increasing its dimension to 100, denoted as ReaRev-5X in the table below. However, even with these modifications, the performance did not match that of NuTrea. To ensure fairness in comparison, all models, including NuTrea, were trained again under identical environments and conditions.
> |  | # layers |  H@1 |  F1  |   |
> |-----------------|:--------:|:----:|:----:|---|
> | ReaRev-2 (base) | 2 | 75.4 | 70.4 | |
> | ReaRev-3|  3 | 74.0 | 69.9 | |
> | ReaRev-4| 4| 73.8 | 69.5 | |
> | ReaRev-5|5| 74.4 | 70.5 | |
> | ReaRev-5X|5  | 74.9 | 70.2 | |
> | NuTrea (ours)|2  | **77.3** | **72.2** | |
>
> &nbsp;
>
> ### Q3. How is the Backup step different from a bigger expansion set in the previous works?
> Compared to previous message passing methods, our Backup’s max-pooling operator (Eq. (7) of manuscript) has its unique advantage in aggregating relevant information from the bag of relations within the multi-hop neighbors (i.e., the subtree). Backup’s effectiveness has been thoroughly demonstrated in our ablation studies (Table 3 of manuscript), providing 3.4 H@1 and 0.6 F1 improvements from the base setting. Furthermore, the experimental results in Q2 reveals that ReaRev with a greater expansion set does not necessarily improve performance.
>
> &nbsp;
>
> ### Q4. Will NuTrea be extremely inefficient in 4 or 5 hop QA?
> Each NuTrea layer consists of three steps: (1) Expansion, (2) Backup, and (3) Node ranking. As the model complexity scales linearly with the number of layers (or hops), similar to GNNs, handling 4 or 5-hop QA settings by adding a few more NuTrea layers will not impact efficiency significantly. NuTrea remains sufficiently scalable for such scenarios.
>
> &nbsp;
>
> ### Q5. RF-IEF was not tested in other models.
> In fact, a plug-in experiment on ReaRev is already provided in Table 4 of Appendix I. We brought the table below for your convenience. We observed that ReaRev’s average performance improves with RF-IEF. Note, the values in parentheses are standard deviations across 5 runs.
> | Models         |   ReaRev   | NuTrea (Ours) |
> |----------------|:----------:|:-------------:|
> | without RF-IEF | 74.2 (1.4) |   75.5 (1.1)  |
> | with RF-IEF| 74.6 (0.8) |   76.3 (1.4)  |
>
> &nbsp;
>
> ### Q6. Bad writing - Lots of forward references to future text.
> We’ll polish our writing and reduce forward references in our final version. Thank you.
>
> &nbsp;
>
> ### Q7. IG model descriptions are very hand-wavy.
> The Instruction Generator (IG) is a commonly used text processing module used in many previous KGQA works, including NSM and ReaRev, and its details have already been described in Appendix B. To further illustrate its details, an IG takes the natural language question $x_q$ as input to output question representations $\\{\mathbf{q}_\text{exp}^{(i)}\\}\_{i=1}^N$ as
>
> $\\{\mathbf{q}\_\text{exp}^{(i)}\\}\_{i=1}^N = \text{IG}_\text{exp}(x_q).$
>
> In the IG function, a tokenizer converts input $x_q$ to tokens $\\{\mathbf{x}\_t\\}\_{t=1}^T$ that are used to retrieve the sentence embedding $\mathbf{q}_\text{LM}$ with a language model $\text{LM}(\cdot)$ (e.g., SentenceBERT) as
>
> $\mathbf{q}\_\text{LM} = \text{LM}(\\{\mathbf{x}\_t\\}\_{t=1}^T).$
>
> To deterministically sample a sequence of sentence representations, a quasi-monte carlo sampling (or non-probability sampling) approach is adopted. $\mathbf{q}_\text{LM}$ is first used to compute attention weights $a_t^{(i)}$ as
>
> $\mathbf{q}^{(i)} = \boldsymbol{W}^{(i)} (\mathbf{q}^{(i-1)} ||   \mathbf{q}\_\text{LM}  ||   \mathbf{q}\_\text{LM} - \mathbf{q}^{(i-1)} ||  \mathbf{q}\_\text{LM} \odot \mathbf{q}^{(i-1)})$\
> $a_t^{(i)} = \text{Softmax}(\boldsymbol{W}_a (\mathbf{q}^{(i)} \odot \mathbf{x}_t)),$
>
> where  $i \in [1,N]$, and  $\mathbf{q}^{(0)}$ is a zero vector. Also, $||$ indicates the concatenation operator, and $\boldsymbol{W}\_a \in \mathbb{R}^{D\times D}$, $\boldsymbol{W}^{(i)} \in \mathbb{R}^{D \times 4D}$ are learnable matrices. Finally, each question representation $\mathbf{q}\_\text{exp}^{(i)}$ is computed as
>
> $\mathbf{q}\_\text{exp}^{(i)} = \sum_t a_t^{(i)} \mathbf{x}_t.$
>
> The same process is repeated to compute the Backup instructions. We will add this enhanced explanation in our final version.
>
> &nbsp;
>
> ### Q8. Are relation and relation type the same thing?
> The term “relation” refers to the edge or link between two entities in a KG, representing a factual connection between them. On the other hand, “relation type” refers to the type or category of the edge (link).
>
> &nbsp;
>
> ### Q9. Can you list situations where Backup is not useful? Or is it always useful?
> Backup is less useful when the global context is less important. Particularly, if the question does not incorporate any conditions or constraints that require considering the broader context of the graph, the Backup module might become redundant. Additionally, if the answer nodes can be easily reached within a few hops from the starting node, the Expansion module alone might be enough to find the correct answer node.

---

> > ### Comment · Reviewer_E45E · 2023-08-12
> >
> > - Most Answers are convincing.
> > -  RFIEF as a generalized method: Marginal gain for reared. Not convinced.
> > - Rearev at higher depth doesn't perform well. This likely means that gains in nutrea are from parameter increase and not from the actual backdrop technique.
> >
> > No Rating change

---

> > > ### Author Response · Authors · 2023-08-13
> > > **Additional Comment by the Authors**
> > >
> > > We appreciate your time in going over our responses! We are glad that you found most of our answers convincing. To resolve your remaining concerns, we have added a few more lines below.
> > >
> > > With regard to your concerns for marginal gains, it is worth noting that RF-IEF was seamlessly integrated into ReaRev, **maintaining the original model hyperparameters to ensure a fair comparison with the base model**. However, ReaRev + RF-IEF may favor a different model architecture (e.g. fewer layers or iterations), and selecting a different hyperparameter set could have further improved its performances.
> > >
> > > Furthermore, we believe the underperformance of the deeper ReaRev models does not necessarily imply that NuTrea took advantage of the larger parameter size. **Our experiment in Q2, in fact, demonstrates that adding more ReaRev layers (i.e., hops) cannot substitute for our NuTrea’s Backup module.** This actually highlights the effectiveness of our approach! Notably, the ReaRev-5X model in the table was trained with the same entity embedding dimension as NuTrea. Even with the same entity embedding size, NuTrea outperformed ReaRev-5X by a significant scale of  2.4 H@1 and 2.0 F1 scores.

---

> > > > ### Comment · Reviewer_E45E · 2023-08-15
> > > >
> > > > Correct me if I'm wrong but Rearev's number of parameters doesn't change with increase of layers.  It might just be that Nutrea doing better than ReaRev is because of higher number of parameters.
> > > >
> > > > Further RFIEF is not a generalized method.
> > > >
> > > > No Score Change.

---

> > > > > ### Author Response · Authors · 2023-08-15
> > > > > **Additional Comment by the Authors**
> > > > >
> > > > > ReaRev's number of parameters _does_ in fact change with the number of layers, as increasing it would influence the number of learnable message-passing operators in the model.
> > > > >
> > > > > What you might be referring to is the "Adaptive Stages" in ReaRev, which basically iterates the entire forward step multiple times to output final predictions -- so, the number of "Adaptive Stages" will not affect the number of parameters. Our NuTrea adopts this tactic as well, which has been briefly discussed in line 14~17 in the supplementary material. Notably, our NuTrea does not require more iterations (i.e. "Adaptive Stages") than ReaRev, and in fact requires only 2 iterations while ReaRev uses 3 on the WebQuestionsSP dataset.

---

> > > > > > ### Comment · Reviewer_E45E · 2023-08-15
> > > > > >
> > > > > > From your intuition about Nutrea, you are capturing larger depth information. Rearev at larger layers is also doing the same. Do you have any intuition on why this is happening? Can you check how many parameters ReaRev-5, ReaRev-5X have?

---

> > > > > > > ### Author Response · Authors · 2023-08-16
> > > > > > > **Additional Comment by the Authors**
> > > > > > >
> > > > > > > ### 1. Can you check how many parameters ReaRev-5, ReaRev-5X have?
> > > > > > > First of all, we would like to inform you that there has been an error in the parameter counter code. To be specific, the original code counted twice the shared Language Model parameters. Hence, we renew the parameter counts as follows.
> > > > > > >
> > > > > > > |                 | # layers | # params | H@1  | F1   |
> > > > > > > |-----------------|----------|----------|------|------|
> > > > > > > | ReaRev-2 (base) | 2        | 23.47 M  | 75.4 | 70.4 |
> > > > > > > | ReaRev-5        | 5        | 23.51 M  | 74.4 | 70.5 |
> > > > > > > | ReaRev-5X       | 5        | 24.72 M  | 74.9 | 70.2 |
> > > > > > > | NuTrea (ours)          | 2        | 27.43 M  | **77.3** | **72.2** |
> > > > > > >
> > > > > > > In the table, our NuTrea has comparable or slightly more parameters than all the ReaRev variants. However, we believe that the increase in parameter size is relatively small considering NuTrea’s huge improvements. Also, ReaRev-5x with 1.25M additional parameters and 3 more layers showed slight performance degradation compared to ReaRev-2 (base). So, we believe that increasing the model size of ReaRev will not provide a significant gain as ours.
> > > > > > >
> > > > > > > To further support our claims, we will try to provide a result with a bigger ReaRev model within the discussion period. We apologize for the confusion caused by the previous response, and thanks again for your questions.
> > > > > > >
> > > > > > > &nbsp;
> > > > > > >
> > > > > > > ### 2. Intuition on the difference between NuTrea and deeper ReaRev
> > > > > > > The key difference is the smoothing effect; NuTrea has a smaller smoothing effect than deeper ReaRev. Assume, for instance, that the model is at a node that is 2 hops away from the seed node (i.e., on the 2nd layer) and needs to be confirmed by the depth-4 information. NuTrea requires only 2 expansion steps (message-passing layers), and 2 backup steps without any node/edge updates. ReaRev, on the other hand, will need up to 6 layers (i.e., hops) to achieve the same effect. Thus, we expect ReaRev to have a greater smoothing effect, which may result in performance degradation.

---

> > > > > > > > ### Author Response · Authors · 2023-08-16
> > > > > > > > **Table Update**
> > > > > > > >
> > > > > > > > As we promised, we are updating the table above in our former response. In the fourth row, we have included ReaRev-5XL, which comes with even more parameters than NuTrea. Its entity dimension has been expanded to 200. However, despite having more parameters, ReaRev-5XL couldn't quite match the performance of NuTrea. These results lead us to conclude that NuTrea holds a clear advantage over the base models.
> > > > > > > >
> > > > > > > > |                 | # layers | # params | H@1  | F1   |
> > > > > > > > |-----------------|----------|----------|------|------|
> > > > > > > > | ReaRev-2 (base) | 2        | 23.47 M  | 75.4 | 70.4 |
> > > > > > > > | ReaRev-5        | 5        | 23.51 M  | 74.4 | 70.5 |
> > > > > > > > | ReaRev-5X       | 5        | 24.72 M  | 74.9 | 70.2 |
> > > > > > > > | **ReaRev-5XL**       | 5        | 27.86 M  | 75.0 | 70.3 |
> > > > > > > > | NuTrea (ours)          | 2        | 27.43 M  | **77.3** | **72.2** |

---

> > > > > > > > ### Comment · Reviewer_E45E · 2023-08-17
> > > > > > > >
> > > > > > > > hey!! thanks for the response.
> > > > > > > > Response to 1: Looks good. There is still an increase in parameters. performance gain vs parameter increase is a subject of discussion but I am not convinced until I can see the results with equal number of parameters.
> > > > > > > >
> > > > > > > > Response to 2: Rearev is a GNN right? so it would still require 4 layers? Elaborate why you need 6.
> > > > > > > >
> > > > > > > >
> > > > > > > > General Response: RFIEF is not justified very well. Parameter increase vs architectural benefits is not well justified. A publication in general is only as strong as its weakest link.
> > > > > > > >
> > > > > > > > No Change in Score

---

> > > > > > > > > ### Author Response · Authors · 2023-08-17
> > > > > > > > > **Additional Comment by the Authors**
> > > > > > > > >
> > > > > > > > > First of all, thank you so much for your active discussions - really appreciate it.
> > > > > > > > >
> > > > > > > > > &nbsp;
> > > > > > > > >
> > > > > > > > > ### 1. I am not convinced until I can see the results with equal number of parameters.
> > > > > > > > > In fact, we have already added a relevant experiment in the above comment "Table Update"! It includes ReaRev-5XL, a model variant with more parameters than our NuTrea. You may see that its performance is not notably better than ReaRev-5X, and that NuTrea outperforms it with significant margins of 2.3 H@1 and 1.9 F1 scores.
> > > > > > > > >
> > > > > > > > > &nbsp;
> > > > > > > > >
> > > > > > > > > ### 2. Rearev is a GNN right? so it would still require 4 layers? Elaborate why you need 6.
> > > > > > > > > ReaRev is indeed a GNN in that it adopts a message passing scheme. But it differs from standard GNN models in that the nodes are not all simultaneously updated. Starting from the seed node, it gradually expands the search area by passing messages one hop at a time, which is basically identical to NuTrea's Expansion step. Then, for an arbitrary node $X_2$ that is 2 hops away from the seed node to access depth-4 information, it will need the message to first arrive at depth-4. But the message will also need to return to $X_2$ to be processed with all the required information. Therefore, the 2 hops from the seed node *PLUS* the way towards depth-4 ($X_2 \rightarrow X_3 \rightarrow X_4$) *PLUS* the way back to node  $X_2$ ($X_4 \rightarrow X_3 \rightarrow X_2$) results in a total of 6 hops (i.e., 6 ReaRev layers). Our NuTrea, on the other hand, can access depth-4 information at $X_2$ very efficiently via the max-pooling scheme in our Backup module.

---

> > > > > > > > > > ### Comment · Reviewer_E45E · 2023-08-18
> > > > > > > > > >
> > > > > > > > > > Thanks for all the clarification. I'm increasing the score. Please address all of my concerns in the final draft if the the paper is accepted.

---

> > > > > > > > > > > ### Author Response · Authors · 2023-08-19
> > > > > > > > > > > **Additional Comment by the Authors**
> > > > > > > > > > >
> > > > > > > > > > > Thank you so much for your support! We will make sure to update our manuscript in our final version.

---

### Official Review · Reviewer_jrbk · 2023-07-07

**Soundness:** 3 good
**Presentation:** 3 good
**Contribution:** 2 fair
**Rating:** 4
**Confidence:** 4

**Summary:**

This paper proposed NuTrea, a graph neural network (GNN) model for Multi-hop KGQA. NuTrea considers broader KG contexts using a tree search scheme to find paths to answer nodes. It uses message-passing layers to explicitly consider question constraints involving bi-directional information (or future context). Also, this paper proposed an interesting node embedding method RF-IEF to characterize KG node entities.

**Strengths:**

- The idea of predicting entities by bi-directional information is reasonable.

- The Relation Frequency–Inverse Entity Frequency metric brined new insight into initializing embeddings of graph nodes.

- The experiments demonstrate the effectiveness of the proposed method.

- The paper is well-written.

**Weaknesses:**

- The core technical design of this work involves utilizing a backup mechanism that allows models to leverage bidirectional information (or global context), enabling them to distinguish entities with similar paths. Some embedding-based methods(e.g., EmbedKGQA and TERP) and GNN-based methods (e.g., SQALER and GraftNet) can also utilize bidirectional information, somewhat sharing a similar high-level idea. The unique superiority compared with these methods and the connections to them deserve more discussions and analyses. Otherwise, it is unclear how to position this work in the current research background, making the technical contribution seem incremental.

- Although the main results and ablation study show performance superiority, most further analyses are about qualitative evaluation. Figure 2 is more about a concept-illustration example instead of an expressiveness analysis. Putting it into other sections (e.g., introduction or method) may be better. Related to the first issue, this work needs more experimental design and in-depth analysis to highlight the unique merits of the proposed methods regarding leveraging the global context. For example, I am curious to see if this method can outperform other models when both the correct and incorrect entities have the same path.

After reading the author’s rebuttal, most of my concerns are addressed. I have no objection to accepting this paper if the authors include more discussion about related works exploiting bidirectional information.

**Questions:**

See weaknesses.

**Limitations:**

I have no concerns about the limitations.

---

> ### Author Rebuttal · Authors · 2023-08-06
>
> ### Q1. What makes NuTrea different from previous embedding-based and GNN-based methods?
> While previous methods (e.g., EmbedKGQA, GraftNet, SQALER) _simultaneously_ update embeddings of all the KG nodes, our NuTrea gradually expands the subgraph and _sequentially_ updates nodes from the seed node towards the answer node, similar to a path-search algorithm. The superiority of this approach has been demonstrated in several recent works, such as NSM (WSDM 2021), TERP (COLING 2021) and ReaRev (EMNLP-Findings 2022). We believe that paths provide a more effective representation of question semantics and are better suited to process the KG in alignment with the given question. Our work builds on this recent line of research by incorporating bi-directional information for enhanced path searching.
>
> &nbsp;
>
> ### Q2. Need for analysis to highlight NuTrea’s merit in leveraging global context. For example, can this method outperform other models when both the correct and incorrect entities have the same path?
> As suggested, we analyzed the ratio of questions that have a KG path (i.e., meta path) that leads to both the correct and incorrect node choices. In the WebQuestionsSP dataset that utilizes the Freebase KG [1], approximately 72% of the questions entailed such KG paths. Thus, outperforming other models on this dataset naturally indicates NuTrea’s unique merit in dealing with circumstances where global context should play a critical role. On the other hand, the MetaQA (1, 2, and 3-hop) dataset that adopts the WikiMovies KB [2] only contained 34% of the questions with such a KG path, which possibly explains the relatively smaller gain in performance on this dataset.
>
> [1] Kurt Bollacker, Colin Evans, Praveen Paritosh, Tim Sturge, and Jamie Taylor. Freebase: a collaboratively created graph database for structuring human knowledge. In ACM SIGMOD, 2008.\
> [2] Alexander Miller, Adam Fisch, Jesse Dodge, Amir-Hossein Karimi, Antoine Bordes, and Jason Weston. Key-value memory networks for directly reading documents. In EMNLP, 2016
>
> &nbsp;
>
> ### Q3. Putting Figure 2 in the Introduction or Method section may be better.
> We will reflect your suggestion in our final version. Thank you.

---

> > ### Comment · Reviewer_jrbk · 2023-08-18
> >
> > - SQALER is more like a path-searching algorithm, while TERP is not. I understand that this paper proposes a new method by incorporating bi-directional information to enhance path searching. Since the bi-directional perspective is not new, what I would like to see is more analyses and comparisons with bi-directional methods and what makes NuTrea better.
> > - The response to Q2 is reasonable.

---

> > > ### Author Response · Authors · 2023-08-19
> > > **Additional Comment by the Authors**
> > >
> > > Thank you for your response. The following is what we understand about the most recent works on multi-hop KGQA.
> > >
> > > - TERP
> > >
> > >     TERP proposes to **align** the KG paths with the natural language question via the rotate-and-scale framework. Here, the question’s textual information is encoded with an LM-based Question Encoder, and the KG relation paths are encoded using both the textual information of the relation and the KG embeddings.
> > >
> > >     So yes, you are absolutely correct in that TERP is more of an embedding-based approach than a path-searching method. We just wanted to emphasize that aligning (or searching) paths with respect to the question semantics is key to handling the multi-hop KGQA problem.
> > >
> > > - SQALER
> > >
> > >     SQALER also tries to **align** the question with the KG, by first extracting a coalesced relational representation from it. Then, an “edge-level” model (e.g. GCN) is applied to refine the solution on the original KG. This provides an efficient means to decouple logical reasoning and multi-hop reasoning, thereby creating a scalable framework for multi-hop KGQA.
> > >
> > > - ReaRev
> > >
> > >     ReaRev is more like a path-searching method that tries to **search** its way from the seed node to the answer nodes while considering the question semantics. That is, each reasoning step (or GNN layer) is a 1-hop expansion of the search area, similar to a BFS search on the KG.
> > >
> > >
> > > Our NuTrea extends this path-**search** framework of ReaRev, by enhancing the algorithm with bi-directional information. To our knowledge, this is the **first** attempt in incorporating bi-directional information into KG path-search. Our approach enriches the context the model can leverage in searching its way to the answer nodes, via our efficient module dubbed “Backup”.
> > >
> > > In order to demonstrate the effectiveness of Backup, we compared NuTrea with deeper versions of ReaRev, which, in theory, should be able to cover the bi-directional information via more search hops (i.e., layers).
> > >
> > > |                 | # layers |  H@1 |  F1  |
> > > |-----------------|:--------:|:----:|:----:|
> > > | ReaRev-2 (base) |     2    | 75.4 | 70.4 |
> > > | ReaRev-3        |     3    | 74.0 | 69.9 |
> > > | ReaRev-4        |     4    | 73.8 | 69.5 |
> > > | ReaRev-5        |     5    | 74.4 | 70.5 |
> > > | NuTrea (ours)         |     2    | **77.3** | **72.2** |
> > >
> > > In the table, various depths were tested, ranging from 2 to 5. However, the model performances did not improve with deeper models. This, in retrospect, indicates that our NuTrea’s Backup module is both effective and efficient in handling the bi-directional information for path searching. We will add these discussions in our final version.

---

### Author Rebuttal · Authors · 2023-08-06

We thank all five reviewers for their strong support and constructive comments on our work. We are glad that the reviewers found our work promising and interesting. Our responses for all the reviewers' questions are provided below. Please go over our responses and let us know if there are issues that are yet unresolved. Attached is a PDF file containing the figure that answers Q4 of reviewer 5UYV.

---

> ### Author Response · Authors · 2023-08-16
> **A Gentle Reminder for Reviewers**
>
> Dear reviewers,
>
> We appreciate your time and effort in reviewing our paper! We have responded to all the reviewers' questions in the comments below. Please go over our responses and let us know if there are any further questions or concerns. Thank you very much!

---

### Decision · Program_Chairs · 2023-09-21

**Decision:**

Accept (poster)

**Comment:**

After extensive discussion between authors and reviewers
about the core contributions of the paper, most of the
reviewers' concerns were addressed by the authors. All
reviewers reflected this in their rating -- except for
reviewer jrbk (borderline reject) who states that they are
fine with accepting the paper if the authors incorporate the
discussion on openreview into the camera ready.

In making a decision on this paper, I see the main issue as
the evaluation of the backup mechanism. Most of the
discussion was about this mechanism and the relevant
baselines.

Backup allows models to leverage bidirectional information
(or global context), enabling them to distinguish entities
with similar paths. There is prior work that utilizes
bidirectional information, i.e., prior work with a similar
high-level idea.  However, what seems to be novel about this
submission is the way that bi-directional information is
incorporated for enhanced path searching (by gradually
expanding the subgraph and sequentially updating nodes).
The intuition is that paths are an effective representation
of the semantics of the question and process the KG in
alignment with the question.  This makes enriched context
available during the search for the answer nodes.

The authors helpfully explain in which
scenarios their proposed method is unlikely to result
in improvements: "easier" questions that can be answered
without taking into account broader graph context and/or
that are located "close" to the starting node.

The authors also provide extensive discussion of ReaRev and
quite a few experimental results for the comparison between
ReaRev and their method.  This additional material persuaded
me that the proposed method is novel compared to
prior work.